# Mapping odorant sensitivities reveals a sparse but structured representation of olfactory chemical space by sensory input to the mouse olfactory bulb

Shawn D Burton[1], Audrey Brown[1], Thomas P Eiting[1], Isaac A Youngstrom[1], Thomas C Rust[1], Michael Schmuker[2], Matt Wachowiak[1]*

[1]Department of Neurobiology, University of Utah School of Medicine, Salt Lake City, United States; [2]Biocomputation Group, Centre of Data Innovation Research, Department of Computer Science, University of Hertfordshire, Hertfordshire, United Kingdom

*For correspondence:
matt.wachowiak@utah.edu

Competing interest: The authors declare that no competing interests exist.

**Abstract** In olfactory systems, convergence of sensory neurons onto glomeruli generates a map of odorant receptor identity. How glomerular maps relate to sensory space remains unclear. We sought to better characterize this relationship in the mouse olfactory system by defining glomeruli in terms of the odorants to which they are most sensitive. Using high-throughput odorant delivery and ultrasensitive imaging of sensory inputs, we imaged responses to 185 odorants presented at concentrations determined to activate only one or a few glomeruli across the dorsal olfactory bulb. The resulting datasets defined the tuning properties of glomeruli - and, by inference, their cognate odorant receptors - in a low-concentration regime, and yielded consensus maps of glomerular sensitivity across a wide range of chemical space. Glomeruli were extremely narrowly tuned, with ~25% responding to only one odorant, and extremely sensitive, responding to their effective odorants at sub-picomolar to nanomolar concentrations. Such narrow tuning in this concentration regime allowed for reliable functional identification of many glomeruli based on a single diagnostic odorant. At the same time, the response spectra of glomeruli responding to multiple odorants was best predicted by straightforward odorant structural features, and glomeruli sensitive to distinct odorants with common structural features were spatially clustered. These results define an underlying structure to the primary representation of sensory space by the mouse olfactory system.

## Editor's evaluation

This paper investigates how odors are represented in the olfactory bulb of the brain. Classical studies have revealed a 'combinatorial code' for odorant recognition, with individual odorants represented by combinations of broadly tuned and low affinity olfactory receptors. Here, the authors perform a large scale analysis of odor responses across glomeruli, and surprisingly observe that odorant receptors instead generally display remarkably narrow tuning profiles.

## Introduction

Across animals, the first central representation of olfactory stimuli arises from the convergence of olfactory sensory neurons (OSNs) that express the same odorant receptor (OR) onto glomeruli of the olfactory bulb (OB) or antennal lobe, generating a map of OR identity across glomeruli. Characterizing odorant responses at the level of OSN input to glomeruli thus enables probing the functional

properties of ORs in the intact animal, as well as understanding of how the representation of olfactory information is structured across the OSN population prior to its processing by central circuits. In the fly olfactory system, comprehensive characterization of OR-defined OSNs mapped to their cognate glomeruli has been foundational for understanding how olfactory information is represented and transformed by successive stages of central processing (*Fishilevich and Vosshall, 2005*; *Hallem and Carlson, 2006*; *Caron et al., 2013*). Achieving a similar level of characterization has been difficult in the mammalian olfactory system: to date, only 3–5% of mammalian ORs have been functionally characterized and mapped to their cognate glomeruli in vivo (*Peterlin et al., 2014*; *Shirasu et al., 2014*; *Saito et al., 2017*).

An additional challenge in the study of olfaction is the complexity of olfactory stimulus space, which includes a large number of compounds ($>>10^4$ -$>>10^9$; *Mayhew et al., 2022*) that are not easily organized along physical dimensions such as wavelength or frequency. A related confound is the strong dependence of OSN specificity on odorant concentration and a lack of consensus on meaningful concentration ranges at which to characterize odor coding strategies (*Meister and Bonhoeffer, 2001*; *Wachowiak and Cohen, 2001*). With few exceptions (*Si et al., 2019*) previous studies have characterized OSN or glomerular responses to odorants presented at one or a few concentrations, typically far above threshold for evoking neural activity (*Rubin and Katz, 1999*; *Wachowiak and Cohen, 2001*; *Nara et al., 2011*; *Ma et al., 2012*; *Chae et al., 2019*; *Pashkovski et al., 2020*; *Soelter et al., 2020*). The consensus from these and other studies is that odorant identity is encoded by combinatorial patterns of OSN and glomerular activity; however, details of such a coding strategy, including the logic of OSN tuning properties, the nature of glomerular maps across the OB surface, and the dimensionality of odorant representations remain unclear.

A useful approach to characterizing sensory response properties and central representations in other sensory systems is to measure neural responses relative to the parts of sensory space to which they are most sensitive – for example, characteristic frequencies in the auditory system (*Evans et al., 1965*). This approach avoids confounds from arbitrarily-chosen stimulus intensities, facilitates comparison across levels and approaches, and can more clearly reveal organizational features such as topographic mapping across neural space and transformations by neural circuits (*Goldstein et al., 1970*; *Stiebler et al., 1997*; *Kandler et al., 2009*). In olfaction, high-affinity odorant-glomerulus or odorant-OR interactions have been identified for only a handful of glomeruli or ORs (*Oka et al., 2006*; *Zhang et al., 2012*; *Zhang et al., 2013*; *Peterlin et al., 2014*; *Sato et al., 2016*; *Horio et al., 2019*; *Soelter et al., 2020*).

Here, we sought to identify 'primary' odorants (i.e. the odorant or odorants to which a glomerulus is most sensitive) for glomeruli of the dorsal OB, using ultrasensitive mapping of OSN inputs to glomeruli in anesthetized mice and efficient screening of a large, chemically diverse panel of odorants. We imaged glomerular responses to 185 odorants in single preparations and, for each odorant, determined a concentration that evoked activation of a small number of glomeruli, thus defining primary odorants for the majority of glomeruli across the dorsal OB. This approach yielded several foundational datasets, including: (1) consensus maps of glomerular odorant representations in the low (sub-nanomolar) concentration range, (2) an atlas of glomerular sensitivities for the dorsal OB across many odorants, and (3) a set of approximately two-dozen individual glomeruli that are robustly identified across animals using their activation by a single diagnostic odorant-concentration combination. These datasets revealed that OSN inputs to OB glomeruli – and, by extension, their cognate ORs – are exquisitely sensitive and selective to their primary odorants, such that representations of olfactory sensory space in this concentration regime are sparse and high-dimensional. Further, co-tuning of glomeruli to their few high-sensitivity odorants, as well as spatial maps of odorant sensitivities, revealed an underlying structure to these sparse representations that reflected relatively straightforward physicochemical features of odorants. This sparse but structured organization identifies an accessible framework for further analyses of how sensory information is represented and processed in in the mammalian olfactory system.

## Results

### Generating consensus maps of odorant sensitivity across dorsal OB glomeruli

To map OSN inputs to OB glomeruli with high sensitivity and consistency across animals, we used tetracycline transactivator-amplified expression of the $Ca^{2+}$ reporter GCaMP6s in all mature OSNs (OMP-IRES-tTA; tetO-GCaMP6s mice; see Materials and methods). Consistent with earlier reports using the OMP-IRES-tTA driver line (*Ma et al., 2014*; *Inagaki et al., 2020*), this expression strategy did not appear to affect targeting of OSNs to their cognate glomeruli (*Zhu et al., 2021*). Odorant-evoked GCaMP6s signals were imaged with widefield epifluorescence across the dorsal surface of both OBs simultaneously in anesthetized mice, using artificial inhalation to ensure consistent odorant sampling (*Eiting and Wachowiak, 2018*).

We used a flexible, high-throughput odorant delivery system (*Burton et al., 2019*) to present a chemically diverse panel of 185 odorants (plus blank and solvent controls) to each experimental preparation. The panel covered a wide range of odorant chemical space as defined by physicochemical descriptors taken from a list of compounds curated for use in flavors and fragrances (*Figure 1A*; *Pashkovski et al., 2020*) (see Materials and methods), and included a diversity of chemical classes as defined by functional group and other structural features (*Supplementary file 1*). Rather than deliver odorants at a single arbitrarily- or empirically-chosen concentration, we used a rational search strategy that allowed adjusting the concentration of each odorant across a >1000 fold (sub-pico- to nanomolar) range in order to identify high-sensitivity odorant-glomerulus interactions and to, ideally, pair each glomerulus with its primary odorant. Guided by recent studies indicating mouse perceptual thresholds in the picomolar range for at least some odorants (*Dewan et al., 2018*; *Williams and Dewan, 2020*), we initially set estimated delivered odorant concentrations to ~1 pM (see Materials and methods). Across a series of pilot experiments, concentrations were then systematically increased (or occasionally decreased) by tenfold steps up to ~1 nM to identify the lowest concentration for each odorant capable of reliably activating at least one glomerulus, with responses averaged across at least three trials (typically, four) per concentration and odorant. Further increases in concentration were not considered as these were unlikely to reveal high-sensitivity interactions. The resulting concentrations were then used to screen the odorant panel across four final mice (eight OBs), with additional variations in concentration tested for many odorants (62–77 per mouse) to achieve comparable activation patterns while accounting for inter-animal variability. Final analyzed concentrations were identical across animals for the majority of odorants (145/185), and none differed by more than tenfold across the four mice.

Nearly all odorants tested proved effective within the picomolar-to-nanomolar concentration range: 163 of 185 odorants (88%) evoked responses in one or more dorsal glomeruli per OB per mouse, and only twelve odorants proved ineffective (i.e. failed to elicit a response in any OB). Response amplitudes were lognormally distributed (Kolmogorov-Smirnov test), with a mean ln(ΔF/F) of 1.48 (corresponding to 4.4% ΔF/F, mean of means across 8 OBs; mean s.d. of responses: 1.03 ln(ΔF/F); number of responses per OB: 308–484).

Qualitatively, patterns of glomerular activation were remarkably consistent across all eight OBs. There were only modest differences in overall response magnitudes across mice or between left and right OBs in the same mouse (ANOVA on all nonzero ΔF/F responses for each OB ($F_{7,3385}$=12.97, $p<2e^{-16}$); post-hoc Tukey's multiple comparisons of means: $p=7e^{-7}$ for 1 of 4 left vs. right OB pairs; $p>0.5$ for 3 of 4 pairs); these differences likely reflect differences in nasal patency between mice or between sides. The cumulative number of glomeruli activated across the odorant panel ranged from 103 to 142 per OB (median: 126) and covered the extent of the dorsal OB (*Figure 1B*; *Figure 1—figure supplement 1*). A recent report mapping ORs to the OB identified ~130 ORs/trace amine-associated receptors (TAARs) from the same area of the OB imaged in the present study (*Zhu et al., 2021*). Allowing for failure to detect low-abundance ORs in *Zhu et al., 2021* and slight mismatches in tissue area, a range of 150–160 glomeruli is a reasonable estimate of the number of glomeruli present in our imaging area. Thus we estimate that our odorant panel was able to identify 75–90% of glomeruli in the imaged area. We used the resulting dataset as a resource for defining consensus high-sensitivity response maps for this large odorant panel across the dorsal OB (*Supplementary file 2*).

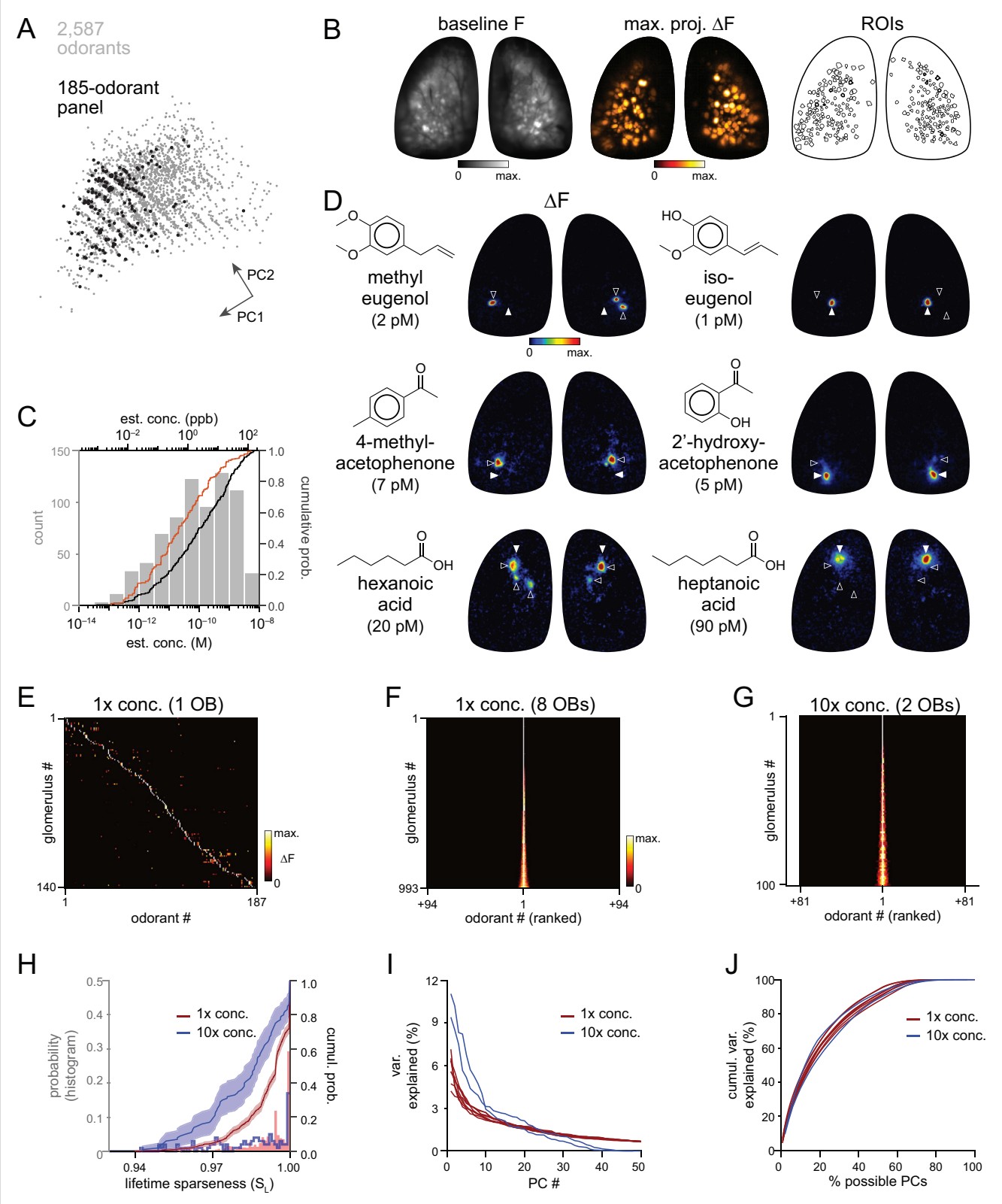

**Figure 1.** High sensitivity and narrow tuning of olfactory sensory input to OB glomeruli. (**A**) Coverage of physicochemical space by the 185-odorant panel. Grey points show the projection of 2,587 odorants across the first two principal components of a matrix of physicochemical descriptors, as in *Pashkovski et al., 2020* (see Materials and methods). Black points indicate the odorants tested in the 185-odorant panel. (**B**) Baseline fluorescence (left), maximal projection of response maps across the 185-odorant panel (middle), and ROIs of responsive glomeruli (right). (**C**) Estimated delivered

*Figure 1 continued on next page*

*Figure 1 continued*

concentrations used across the odorant panel. Histogram and black cumulative distribution function show concentrations of each presented odorant across four preparations (n=740). Red cumulative distribution function shows the minimal effective concentration for each responsive glomerulus (n=993). (**D**) Response maps evoked by single odorants for the preparation shown in (**B**). Each row shows distinct but neighboring glomeruli (demarcated by filled and open arrowheads) activated by structurally similar odorants. Estimated concentrations are rounded to single-significant digit precision. (**E**) Matrix of responses across all responsive glomeruli in one OB. Each row (glomerulus) is normalized to its maximal response across the odorant panel. Glomeruli are sorted in order of their maximally-activating odorant, producing a pseudo-diagonalized matrix. Odorants are ordered according to nominal structural classification (see Materials and methods). Matrix includes responses to empty and solvent controls. (**F, G**) Response spectra of all imaged glomeruli (rows) across the odorant panel (columns), normalized by maximal response, for 1 × concentration epifluorescence dataset and the 10 × concentration two-photon dataset (separate preparations; 10 × two-photon data imaged from a smaller field of view containing fewer glomeruli). Odorant order sorted by response amplitude; glomerular order sorted by lifetime sparseness. (**H**) Histogram and cumulative distribution functions of lifetime sparseness ($S_L$) values for all responsive glomeruli for the odorant panel presented at original, 1 × concentrations (red; n=993 glomeruli) and at 10 × concentrations (blue; n=100). Shading denotes 95% confidence intervals (calculated using 'ecdf' function in Matlab). (**I**) Percent of variance in glomerular responses to the odorant panel explained by each successive PC, plotted for each OB. Red plots: 1 × concentrations, n=8 OBs; blue plots: 10 × concentrations, n=2 OBs. (**J**) Cumulative variance in glomerular responses to the odorant panel explained by increasing fractions of possible PCs (constrained by the number of responsive glomeruli in each OB).

The online version of this article includes the following source data and figure supplement(s) for figure 1:

**Source data 1.** Source data for *Figure 1C*.

**Source data 2.** Source data for *Figure 1H*.

**Figure supplement 1.** Summary of responsive glomeruli and their response spectra.

**Figure supplement 1—source data 1.** Source data for *Figure 1—figure supplement 1*.

**Figure supplement 2.** Comparison of delivered odorant concentrations across studies.

**Figure supplement 2—source data 1.** Source data for *Figure 1—figure supplement 2*.

**Figure supplement 3.** Sparse glomerular responses evoked across the 185-odorant panel.

**Figure supplement 3—source data 1.** Source data for *Figure 1—figure supplement 3*.

**Figure supplement 4.** Two-photon imaging of glomerular odorant responses to tenfold higher odorant concentrations.

## Sensory inputs to glomeruli are highly sensitive and narrowly tuned to their primary odorants

Across all odorants, final estimated concentrations ranged from $4 \times 10^{-14}$ to $4 \times 10^{-9}$ M (median: $1 \times 10^{-10}$ M, 2.4 ppb) (*Figure 1C*; *Supplementary file 1*). Maximal glomerular sensitivity (i.e. the concentration at which each glomerulus responded to its primary odorant) was distributed across even lower concentrations, with a median of $2 \times 10^{-11}$ M (0.5 ppb) (*Figure 1C*). These concentrations are, overall, substantially lower than those used to characterize odorant representations in earlier studies, by as much as 4–5 orders of magnitude (*Figure 1—figure supplement 2*), and indicate that high odorant sensitivity is a general feature of OSN inputs to glomeruli. We next assessed how canonical features of olfactory stimulus coding at the level of OSN input to glomeruli manifest in this concentration regime, beginning with the tuning of individual glomeruli and the nature of glomerular representations of individual odorants.

Our experimental approach was designed to yield sparse glomerular responses to each odorant. Nevertheless, the degree of sparseness was striking, with each effective odorant evoking input to only a few glomeruli (median: 2 glomeruli per odorant; quartiles: 1–3 glomeruli per odorant; 1290 responses across 8 OBs) (*Figure 1—figure supplement 3A*). Population sparseness ($S_P$), a measure of the selectivity of glomerular activation for a given odorant, was exceptionally high (median $S_P$: 0.994; quartiles: 0.990–0.997, calculated from mean $S_P$ per odorant across 8 OBs) (*Figure 1—figure supplement 3B*). There was no difference in $S_P$ between the left and right OBs of the same mouse (paired Wilcoxon Signed-Ranks test, Bonferroni-corrected p-values for each of 4 mice: p=0.12, 1.0, 1.0, 1.0). This result implies that individual glomeruli are narrowly tuned across the entire odorant panel. Indeed, highly selective tuning was evident when comparing odorant response maps, with individual glomeruli often responding strongly to a given odorant and not at all to structurally similar odorants (*Figure 1D*).

Earlier characterizations of OR tuning have identified a mix of narrowly- and broadly-tuned ORs (*Hallem and Carlson, 2006*; *Saito et al., 2009*; *Nara et al., 2011*). Here, we observed narrow tuning of OSN inputs to nearly all glomeruli, and for glomeruli tuned to nearly all odorants (*Figure 1E and*

*F*; *Figure 1—figure supplement 3C*). Lifetime sparseness ($S_L$), a measure of stimulus selectivity that reflects the distribution of response magnitudes across a stimulus set and ranges from 0 (no stimulus selectivity) to 1 (response to only a single stimulus) (*Davison and Katz, 2007*; *Schlief and Wilson, 2007*), was extremely high across the glomerular population, with a median value of 0.995 (mean of medians across 8 OBs; mean quartiles: 0.989–1) (*Figure 1H*). 25–30% of glomeruli in a given OB (mean ± s.d.: 29% ± 6%, 288/1004 total glomeruli) responded to only one odorant from the panel, and only 19% of glomeruli (187/1004) responded to more than five odorants (*Figure 1—figure supplement 3C*).

Sparse responses and narrow tuning of glomeruli was not due to an 'iceberg effect', in which only a few odorants evoked responses detectable above the signal-to-noise ratio of our imaging approach, as response amplitudes were typically many times greater than baseline variance levels. 75% of glomeruli that responded to more than one odorant (n=716) had a dynamic range (i.e. the ratio between the strongest and weakest odorant responses) greater than 2, 50% had a range greater than 4, and 20% had a range greater than 10, indicating a high dynamic range in odorant responsiveness even in this low-concentration regime. $S_L$ values were also consistently high and independent of both dynamic range and maximum response amplitude (*Figure 1—figure supplement 3D, E*), indicating that sparse responses are not an artifact of low signal-to-noise ratios or of weakly-responding glomeruli.

We further tested the concentration-dependence of OSN tuning by presenting 159 odorants of the panel at tenfold higher concentrations in two separate preparations (the remaining 26 odorants were presented at the same concentration as the widefield imaging set but omitted from this concentration-dependent analysis; see *Supplementary file 1*), using two-photon imaging from a central subregion of the dorsal OB in order to more precisely attribute signals to specific glomeruli (*Figure 1—figure supplement 4*). Glomerular tuning was only slightly broader at these higher concentrations, with 12% of glomeruli activated by only one of the 159 odorants and a median $S_L$ of 0.983 (quartiles: 0.973–0.994; 100 responsive glomeruli, 2 mice) (*Figure 1G and H*). Narrow OSN tuning was thus not restricted to carefully-chosen perithreshold concentrations, suggesting that sparse and selective glomerular activation is a robust feature of odorant representations in low-concentration regimes.

Previous studies examining neural responses to higher concentrations have proposed that odorant identity can be reliably encoded within a structured, lower-dimensional neural coding space (*Chae et al., 2019*; *Si et al., 2019*; *Pashkovski et al., 2020*). However, glomerulus-odorant response matrices in our low-concentration datasets appeared high-dimensional, at least in the Euclidean domain. The first principal component (PC) in each OB dataset accounted for only 4–7% of the total variance (*Figure 1I*), and 54–67 PCs (44–55% of possible PCs, given 100–140 glomeruli per OB) were required to account for 90% of the response variance (*Figure 1J*), with similar results observed for the tenfold higher concentration two-photon dataset. These results substantially differ from the 21 PCs required to account for 90% of the response variance in a pseudopopulation of 871 glomeruli (i.e. ~2.4% of possible PCs) in a comparable analysis of glomerular responses to substantially higher odorant concentrations (*Chae et al., 2019*). Effective dimensionality (ED), a measure of response covariance across a dataset (*Litwin-Kumar et al., 2017*), was also substantially higher in our main dataset (mean ± s.d.: 48.6±4.4, n=8 OBs) than that reported for odorant responses imaged from OB projections to piriform cortex using higher odorant concentrations (ED: 13.5, n=22 odorants x 3160 axonal boutons) (*Pashkovski et al., 2020*).

Ideally, it would be useful to compare glomerular tuning and odorant representations for the low-concentration ranges tested here to the tuning and representations observed in comparable earlier studies using the exact same odorants. To that end, we took advantage of a publicly available dataset describing OSN inputs to mouse OB glomeruli, collected using near-identical imaging methods and the same OMP-IRES-tTA driver line as in the present study, but with a less-sensitive GCaMP2 reporter and higher odorant concentrations (*Ma et al., 2012*). We generated response matrices using the lowest effective concentrations reported in this previous study for the 31 odorants common with the current study (median: 4.3 ppm; quartiles: 1.0–15.1 ppm), which were approximately 1000 x higher than in our dataset (median: 3.4 ppb; quartiles: 0.5–25 ppb) (*Supplementary file 3*). As expected, odorants in this higher-concentration regime activated substantially more glomeruli across the same area of the dorsal OB, with a quartile range of 2–23 glomeruli per odorant (median: 5) compared to a range of 1.4–2.9 in our dataset (median of mean values across 8 OBs: 1.75; p=3 × 10$^{-5}$, paired Wilcoxon Signed Ranks). $S_L$ of individual glomeruli across the 31 odorants was also significantly lower,

with a quartile range of 0.84–0.97 (median: 0.9; n=98 glomeruli from one OB) compared to a range of 0.97–1 in our dataset (median: 1; n=273 glomeruli from 8 OBs; p=0, Mann-Whitney U test). Dimensionality of the glomerular responses across the 31 odorant panel was also substantially lower for the higher concentration data, with an ED of 5.3 compared to 13.2±1.1 (mean ± s.d., 8 OBs). This comparison thus further confirms that the narrow tuning of glomeruli and sparser, higher dimensional odorant response patterns in our dataset directly reflect differences in how odorant information is represented in this much lower concentration regime, rather than differences in odorant panel composition.

## Diagnostic odorants enable widespread functional identification of glomeruli

The narrow tuning and high sensitivity of glomeruli to their primary odorants suggests that many glomeruli can be identified simply by their response to a single diagnostic odorant delivered at low concentration. Examples of five such diagnostic odorants evoking consistent singular- or near-singular activation of glomeruli are shown in *Figure 2A*. These glomeruli appear in a consistent location in each OB and exhibit near-identical response spectra across the full odorant panel (*Figure 2B*). For example, the aromatic ester phenyl acetate (~3 pM) strongly activated a single glomerulus in the central OB, with identical response spectra across the 8 OBs (*Figure 2A and B*). Likewise, the odorant methyl tiglate (~10 pM) strongly activated a single glomerulus in the central-medial OB in each of the 8 OBs imaged; this glomerulus was also strongly- and singly-activated by ethyl, hexyl, and isopropyl tiglate, as well as trans-2-methyl-2-butenal and 2-methyl-2-pentenoic acid (*Figure 2A and B*). These observations suggest that, in many cases, a single diagnostic odorant can be used to identify putatively cognate glomeruli across animals.

To identify additional such glomeruli and their diagnostic odorants from the dataset, we first screened odorants using two highly conservative criteria for sparseness and response reliability. To screen for singular- or near-singular glomerular activation, we required that an odorant activate no more than two glomeruli above a 50% $\Delta F_{max}$. cutoff across all eight imaged OBs. To screen for reliability, we required that an odorant activate at least one glomerulus in at least six of eight OBs, and in each of the four mice. For each of the 80 odorants passing this screen, we tested the odorant's ability to identify glomeruli across OBs by comparing the response spectrum of the glomerulus most strongly activated by that odorant across the full 185-odorant panel with that of all other glomeruli in each OB. For 19 odorants, the strongest (or only) activated glomerulus identified the glomerulus with the highest-correlated response spectrum in 100% of comparisons (median correlation coefficient [Pearson's r] across OBs: 0.95±0.04; mean ± s.d., n=19). Relaxing these criteria slightly to allow for inherent variability in responsiveness across the odorant panel – to a cutoff of 80% match between the strongest-activated and most-correlated glomerulus across OBs (error ratio <0.2; see Materials and methods) and a median correlation coefficient in response spectra of >0.8 – yielded 22 additional odorants. Thus, 41 odorants from the original 185-odorant panel could serve as diagnostic probes to identify 26 unique glomeruli (*Table 1*; *Supplementary file 4*). *Supplementary file 5* lists the remaining odorants that elicited reliably sparse glomerular activity but which did not meet our conservative criteria for functional identification. Notably, while spatial location was not used as a diagnostic criterion, glomeruli identified with this approach appeared in a similar location on the dorsal OB, with a spatial jitter consistent with that characterized for OR-defined OSN projections (*Figure 2C*; *Table 1*; *Zapiec and Mombaerts, 2015*). Diagnostic odorant-concentration pairs also evoked singular activation of the same putatively identified glomeruli in awake, head-fixed mice at the same low concentrations (*Figure 2—figure supplement 1*), suggesting that OSNs are similarly sensitive and selective during natural odorant sampling. Thus, the high selectivity and sensitivity of OSNs for their primary odorants allows for simple and robust identification of cognate glomeruli across OBs using a single odorant-concentration pair.

Singularly-activated glomeruli presumably reflect odorant binding to a single OR species whose sensitivity to that odorant is higher than that of all other dorsally projecting ORs. OSNs expressing most ORs are expected to project to two glomeruli, only one of which would potentially be visible on the dorsal surface, with the other glomerulus located medially and inaccessible to our imaging approach (*Nagao et al., 2000*; *Zhu et al., 2021*). Notably, while a relative few ORs have been mapped to their cognate glomeruli and functionally characterized (*Peterlin et al., 2014*), several of the functionally-identified glomeruli had diagnostic odorants and spatial locations that were a close match

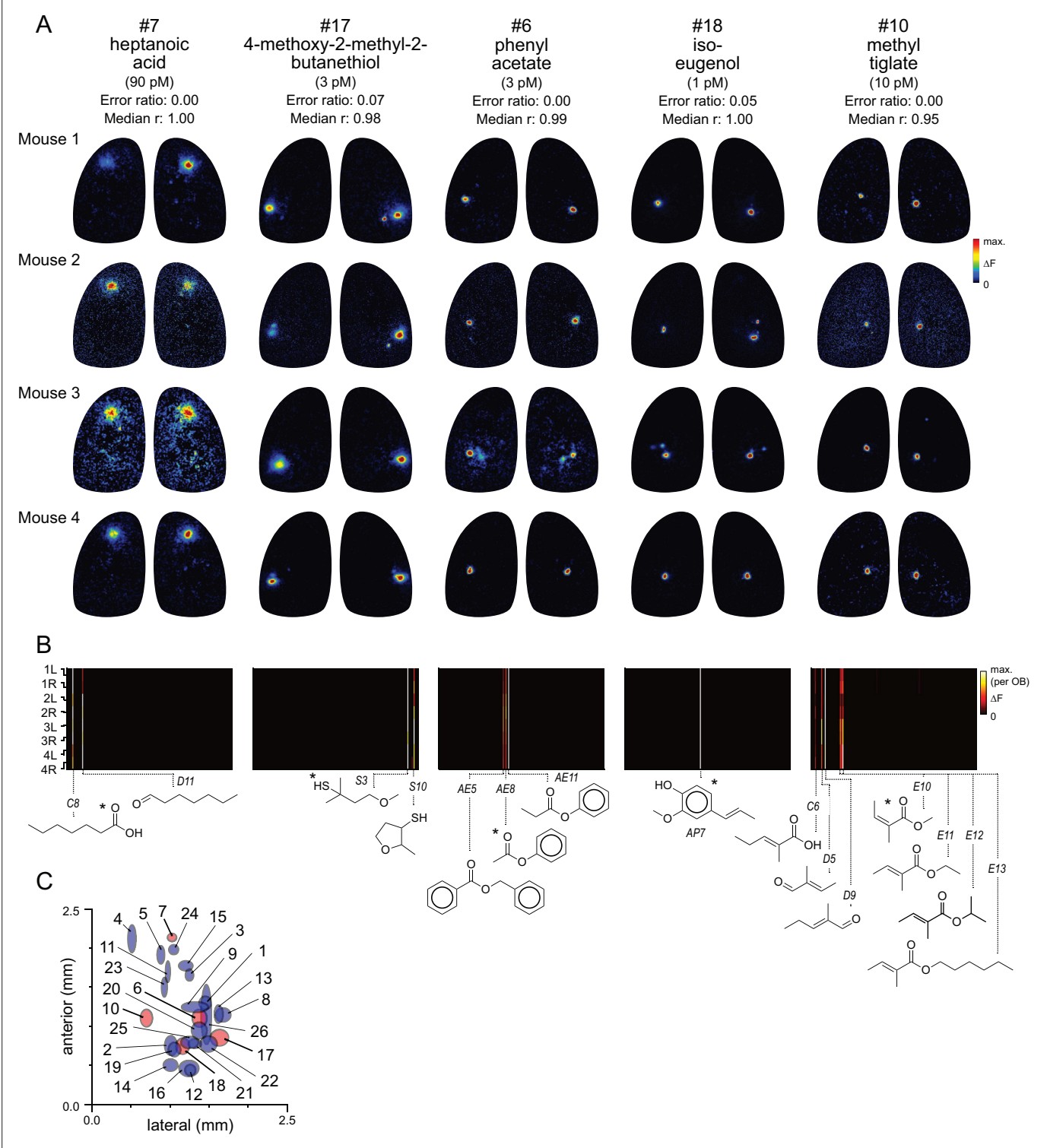

**Figure 2.** Functional identification of glomeruli using singular activation by diagnostic odorants. (**A**) Response maps evoked by five odorants (columns), shown for each of four mice (rows), eliciting singular or near-singular activation of a glomerulus in a consistent location in each OB. See Text for definition of error ratio and median r. Estimated delivered concentrations are rounded to single-significant digit precision. (**B**) Response spectra for each glomerulus in (**A**) across the 185-odorant panel (columns), shown for each of the eight imaged OBs (rows). Pseudocolor scale is normalized to the maximal response in the glomerulus for each preparation. Structures of effective odorants are shown at bottom. Asterisk indicates diagnostic odorant shown in (**A**). Letter-number abbreviations indicate odorant identity, as listed in **Supplementary file 1**. (**C**) Mean locations of all functionally identified

*Figure 2 continued on next page*

*Figure 2 continued*

glomeruli, referenced to the midline and caudal sinus of the OB. Spot width and height indicate jitter (s.d.) of medial-lateral and antero-posterior location across the eight OBs. Identified glomeruli from (**A**) are shown in red; all other glomeruli shown in blue.

The online version of this article includes the following figure supplement(s) for figure 2:

**Figure supplement 1.** Diagnostic odorants functionally identify glomeruli with similar sensitivity in awake mice.

**Figure supplement 2.** Functional identification of putative TAAR-associated glomeruli using amine odorants.

**Figure supplement 3.** Identified glomeruli maintain narrow tuning across a tenfold increase in odorant concentrations.

to OR-defined glomeruli. For example, 2′-hydroxyacetophenone (glom. #12), 2-methylacetophenone (glom. #21) and methyl eugenol (glom. #22) selectively activated glomeruli that closely matched the reported positions of glomeruli for Olfr160 (*Zhang et al., 2012*), Olfr1377 (*Zhu et al., 2021*), and Olfr510 (*Oka et al., 2006*; *Supplementary file 4*). For TAAR-expressing OSNs, both their medial and lateral glomeruli are dorsal, such that activation of at least some TAARs should activate pairs of dorsal glomeruli (*Pacifico et al., 2012*; *Sato et al., 2020*). Indeed, several amine odorants that have been previously identified as preferred ligands for particular TAARs evoked strong but selective activity in pairs of glomeruli within the putative TAAR-projecting domain, including cadaverine, β-phenylethyl-amine, and isopentylamine, preferred ligands for TAAR9, TAAR4, and TAAR5, respectively (*Zhang et al., 2013*; *Saraiva et al., 2016*; *Figure 2—figure supplement 2*). These paired glomeruli had identical or near-identical response spectra across the full odorant panel, consistent with their repre-senting the same TAAR. Overall, these results suggest that functionally identifying glomeruli from simple diagnostic odorant-concentration pairs may be a useful platform for linking glomeruli to their cognate, genetically defined ORs in vivo.

We used the current set of identified glomeruli to further explore the logic of OSN tuning by analyzing their response spectra, compiled across the eight OBs (*Figure 3A*). Consensus response spectra were defined by the median response to each odorant across the eight OBs. The median $S_L$ across the 26 glomeruli was 0.99 (mean ± s.d.: 0.988±0.011), indicating that the functionally-identified glomeruli exhibited similarly narrow tuning as the general glomerular population. Seven of the 26 glomeruli responded to only a single odorant. Of the remaining 19 glomeruli, effective co-tuned odorants often shared common structural features such as functional group, carbon chain length (for aliphatic odorants), ring structure, or heteroatom. At the same time, nearly all glomeruli were exqui-sitely selective to their co-tuned odorants, often showing a strong response to one and no response to other structurally similar odorants. Several of the identified glomeruli appeared relatively broadly tuned to heterocyclic compounds including pyrazines and thiazoles (e.g. glomeruli 13, 14, and 16, *Supplementary file 4*). In addition, several glomeruli showed high sensitivity to odorants where chemical similarity was less obvious; for example, glomeruli 15, 19, and 20 were sensitive to both aromatic and aliphatic compounds, and glomerulus 14 was highly sensitive to the cyclic terpenoid (R)-(+)-pulegone as well as several pyrazines.

To further test the concentration-dependence of glomerular tuning, we located six of the functionally-identified glomeruli using epifluorescence widefield imaging at their diagnostic ('1 x') odorant concentrations, and then characterized their response spectra to the tenfold-higher concen-tration odorant panel ('10 x') using two-photon imaging (n=1–2 OBs), as described above. Consistent with the summary analysis across all glomeruli, the response spectrum of each glomerulus at these 10 x concentrations was nearly identical to the median response spectrum at the 1 x concentrations, with smaller-magnitude responses occasionally recruited in response to some additional odorants (*Figure 2—figure supplement 3*). This result further supports the conclusion that narrow tuning is a robust feature of OSN inputs to OB glomeruli.

## Basic structural features predict co-tuning of glomeruli to their high-sensitivity odorants

Earlier studies have used sets of physicochemical descriptors of odorants to infer relationships between the chemical space of odorants and their neural representations, OR tuning, or odor percep-tion (*Haddad et al., 2008*; *Saito et al., 2009*; *Chae et al., 2019*; *Soelter et al., 2020*; *Gerkin, 2021*). Given the exceptionally narrow tuning of glomerular inputs in our dataset, we used this approach to explore the chemical relationships between the small number of odorants to which OSN populations

**Table 1.** Diagnostic odorants and concentrations for functionally-identified glomeruli.

Error ratio: Incidence of mismatch between strongest-activated glomeruli and glomeruli with most correlated odorant response spectra (ORS) across 2 OBs, divided by all potential 2-OB comparisons.

Median ORS corr.: Median ORS correlation coefficient (Pearson's r) across all pairwise comparisons of ORS for the maximally-activated glomerulus in each responsive OB.

Mediolateral: Position of glomerulus centroid in the mediolateral axis, in units of μm from the midline (mean ± s.d.).

Anteroposterior: Position of glomerulus centroid in the anterior-posterior axis, in units of μm from the transverse sinus delineating the posterior margin of the OB (mean ± s.d.).

| # | Odorant | Est. conc. (M) | Error ratio | Median ORS corr. | Mediolateral (μm) | Anteroposterior (μm) |
|---|---------|----------------|-------------|------------------|-------------------|----------------------|
| 1 | Benzaldehyde | 8E-11 | 0.00 | 1.00 | 1448.3±76.9 | 1298.4±82.4 |
| 2 | Elemicin | 5E-12 | 0.00 | 1.00 | 1011.6±80.9 | 753.2±126.7 |
| 3 | Vanillin | 3E-11 | 0.07 | 1.00 | 1249.6±54.8 | 1651±75.9 |
| 4 | Trans-2-dodecenal | 8E-10 | 0.00 | 1.00 | 514.9±54.2 | 2117.5±179.5 |
|   | Ethyl phenylacetate | 7E-11 | 0.00 | 1.00 | | |
| 5 | Allyl phenylacetate | 1E-10 | 0.00 | 1.00 | 884±50.5 | 1911.9±118.9 |
|   | Phenyl acetate | 3E-12 | 0.00 | 0.99 | | |
| 6 | Phenyl propionate | 1E-11 | 0.00 | 0.99 | 1375.6±93.3 | 1103.4±111.8 |
|   | Heptanoic acid | 7E-11 | 0.00 | 0.97 | | |
| 7 | Heptanal | 2E-9 | 0.00 | 0.95 | 1026.6±60.5 | 2132.8±54.7 |
| 8 | Methional | 1E-11 | 0.00 | 1.00 | 1680.2±99.6 | 1144.1±91.3 |
| 9 | 3-Mercaptohexyl acetate | 4E-12 | 0.00 | 0.97 | 1316.2±170.4 | 1244.5±67.3 |
|   | Trans-2-methyl-2-butenal | 1E-11 | 0.00 | 0.95 | | |
|   | 2-Methyl-2-pentenal | 6E-11 | 0.00 | 0.95 | | |
|   | Methyl tiglate | 1E-11 | 0.00 | 0.95 | | |
|   | Ethyl tiglate | 2E-12 | 0.00 | 0.95 | | |
|   | Isopropyl tiglate | 2E-11 | 0.00 | 0.95 | | |
| 10 | Hexyl tiglate | 4E-10 | 0.00 | 0.95 | 700.8±79.3 | 1103.7±113.9 |
|   | Isovaleric acid | 9E-12 | 0.00 | 0.92 | | |
| 11 | Isovaleraldehyde | 4E-9 | 0.00 | 0.92 | 973.5±32.4 | 1696.9±138 |
| 12 | 2'-Hydroxyacetophenone | 5E-12 | 0.00 | 0.86 | 1258.5±66.8 | 444±68.9 |
| 13 | Pyrazine | 2E-9 | 0.03 | 0.91 | 1618±54.2 | 1154.8±117.4 |
|   | 2-Isobutyl-3-methoxypyrazine | 3E-11 | 0.04 | 0.91 | | |
| 14 | (R)-(+)-pulegone | 7E-13 | 0.04 | 0.91 | 1007.4±93.9 | 503.6±80.8 |
| 15 | 4-(4-Hydroxyphenyl)–2-butanone | 4E-10 | 0.02 | 0.84 | 1202.5±92.4 | 1771.8±66.1 |
|   | 2,4,5-Trimethylthiazole | 5E-12 | 0.04 | 0.91 | | |
| 16 | Ethyl-2,5-dihydro-4-methylthiazole | 3E-10 | 0.04 | 0.91 | 1242.6±128.3 | 460.8±106.6 |
|   | 4-Methoxy-2-methyl-2-butanethiol | 3E-12 | 0.07 | 0.98 | | |
| 17 | 2-Methyl-3-tetrahydrofuranthiol | 2E-10 | 0.07 | 0.98 | 1633.1±116.8 | 853.6±107.1 |
| 18 | Isoeugenol | 8E-13 | 0.05 | 1.00 | 1162.9±86.7 | 746.2±105.4 |
| 19 | Menthone | 3E-10 | 0.05 | 0.85 | 1058.4±76.9 | 701.8±93.8 |

*Table 1 continued on next page*

Table 1 continued

| # | Odorant | Est. conc. (M) | Error ratio | Median ORS corr. | Mediolateral (µm) | Anteroposterior (µm) |
|---|---------|---------------|-------------|------------------|-------------------|----------------------|
| 20 | 2-Hexanone | 1E-9 | 0.11 | 0.94 | 1372±98.3 | 945.6±105.8 |
| | Acetophenone | 1E-11 | 0.02 | 0.83 | | |
| 21 | 2-Methylacetophenone | 1E-12 | 0.02 | 0.83 | 1317.2±81.1 | 776.9±61.6 |
| 22 | Methyl eugenol | 2E-12 | 0.13 | 0.90 | 1491.6±107 | 771.2±103.5 |
| | 2-Methylbutyraldehyde | 8E-11 | 0.16 | 0.87 | | |
| | 2-Methylvaleraldehyde | 1E-10 | 0.16 | 0.87 | | |
| 23 | Methyl 2-methylbutyrate | 1E-10 | 0.16 | 0.87 | 928.6±38.7 | 1497.3±126.9 |
| 24 | Hexanal | 7E-10 | 0.18 | 0.97 | 1047.7±63.7 | 1977.8±66 |
| 25 | Fenchol | 5E-10 | 0.16 | 0.98 | 1251±109.5 | 795.9±76.9 |
| 26 | 5-Methylfurfural | 5E-10 | 0.19 | 1.00 | 1465±67.1 | 1153.3±380.5 |

were most sensitive. We first focused on the 19 functionally-identified glomeruli that were responsive to more than one odorant, sorting the median response spectra of each glomerulus according to odorant distance in physicochemical descriptor space, relative to the primary odorant for that glomerulus. We initially used a subset of 1377 descriptors from the E-Dragon web app (*Todeschini and Consonni, 2003*; *Tetko et al., 2005*) chosen previously from a large-scale characterization of mammalian OR binding properties (*Haddad et al., 2008*; *Saito et al., 2009*; *Chae et al., 2019*; *Soelter et al., 2020*; *Gerkin, 2021*). This descriptor set appeared moderately effective at predicting glomerulus co-tuning given the primary odorant identity, although some glomeruli were co-tuned to odorants distributed across large distances in the descriptor space, and all glomeruli failed to respond to numerous odorants located more closely within this space (*Figure 3B*).

To quantitatively assess the ability of physicochemical descriptors to predict glomerulus co-tuning, we used a performance metric from receiver-operating characteristic analysis (see Materials and methods), which is commonly used in ligand-based virtual screening for drug discovery (*Duan et al., 2010*; *Lopes et al., 2017*). The metric reflected the cumulative fraction of responses explained by each successively ranked odorant, ranked according to distance in the chemical space defined by a descriptor set, relative to shuffled odorant ordering (*Figure 3C*), and had a value of 1 for perfect prediction of odorant responses and 0 for prediction no different from chance. Using the 1377-element Dragon descriptor set to define the chemical space of our odorant panel, and the primary odorant as the query odorant from which distance in this descriptor-defined space was measured, performance metrics for functionally identified glomeruli were well above chance (median: 0.73; quartiles: 0.60–0.96). Similar results were obtained with models of chemical space defined by expanded physicochemical descriptor sets (2982 descriptors, Alvadesc) (*Pashkovski et al., 2020*) (median: 0.75; quartiles: 0.63–0.93), and when querying response spectra using the strongest-activating (rather than most-sensitive, or primary) odorant as the query odorant (median: 0.82; quartiles: 0.66–0.96) (*Figure 3C*, *Figure 3—figure supplement 1A*).

We next extended this analysis to the full dataset of glomerular responses. Performance metrics were calculated for all glomeruli responding to more than one odorant at a level greater than 10% of its maximal response (n=694 glomeruli). To avoid assuming a priori knowledge of the primary odorant for a glomerulus, for this analysis we measured performance using each effective odorant as the query odorant, then took the median of these values as the performance metric for each glomerulus (see Materials and methods). We also tested additional reductions of chemical space that have been used in prior studies to define odorant similarity based on physicochemical features. Specifically, we used an optimized subset of 32 Dragon descriptors selected from fitting to earlier published odorant response datasets (*Haddad et al., 2008*), a subset (49) of the 154 descriptors defining odorant functional groups (*Saito et al., 2009*), and a subset (108) of the 166 MACCS chemical feature keyset bits that were previously developed for ligand-based virtual screening (*Durant et al., 2002*) (see Materials

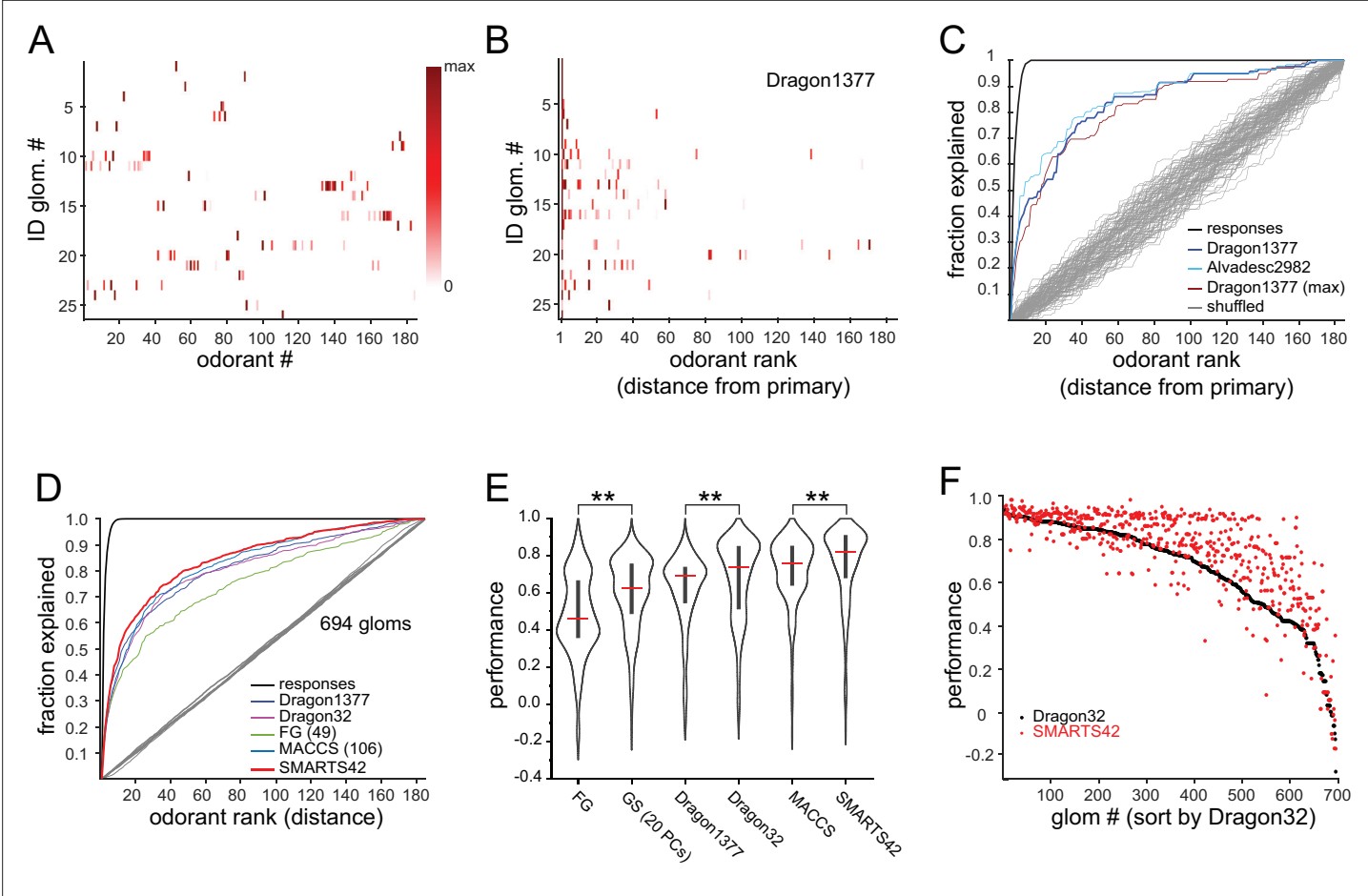

**Figure 3.** Predicting odorant response specificity from physicochemical feature sets. (**A**) Median response spectra of all functionally-identified glomeruli. Each spectrum (row) is normalized to its maximal odorant response. Odorants ordered according to nominal structural classification (see Materials and methods), as in *Figure 1E*. (**B**) Median response spectra of functionally-identified glomeruli, with odorants ordered by physicochemical descriptor distance from the primary odorant for each glomerulus (odorant that evokes a response at the lowest concentration), using the Dragon1377 descriptor set. (**C**) Cumulative fraction of all odorant responses encountered with increasing ranked distance from the primary odorant, averaged across all 19 functionally identified glomeruli with co-tuning (i.e. responding to more than one odorant). Colored lines show the cumulative response fraction observed for different odorant rankings, including distance in: the Dragon1377 descriptor set (as shown in (**B**)), the Dragon1377 descriptor set re-ordered relative to the strongest-activating odorant (Dragon1377 (max)), and a larger 2982-element descriptor set (Alvad. 2982). Black line shows the cumulative response fraction of binarized odorant responses, with odorants ranked by response magnitude, representing the maximum achievable response prediction by odorant ranking. Grey lines show cumulative response fraction following random odorant ranking (100 iterations), representing chance prediction. (**D**) Same as (**C**) for all co-tuned glomeruli imaged across the eight OBs (n=694), relative to the primary odorant for each glomerulus, for different odorant rankings, including distance in: the Dragon1377 descriptor set, a previously optimized subset of 32 Dragon descriptors (Dragon32), a subset of descriptors defining functional groups (FG(49)), a subset of previously developed chemical feature binarized fingerprints (i.e. bits; MACCS(106)), and a novel set of chemical feature binarized fingerprints (SMARTS42). Grey line shows mean cumulative response fraction following random odorant ranking. (**E**) Distribution of performance metrics (see Materials and methods; 1 indicates perfect prediction, 0 indicates chance prediction) across all 694 co-tuned glomeruli for different odorant rankings. Red bar, median; Center bar, interquartile range; envelope, smoothed point density. Asterisks indicate significant difference between odorant rankings (Kruskal-Wallis test comparing all performance metrics: $p<2 \times 10^{-16}$; chi-squared statistic, 859; df = 6; p=0.002 for MACCS(106) vs. Dragon32; $p<1 \times 10^{-25}$ for MACCS(106) vs. all other rankings (post-hoc Dunn tests, Benjamani-Hochberg correction for multiple comparisons); only differences between incrementally higher medians are shown for clarity). (**F**) Prediction performance metrics for each of the 694 co-tuned glomeruli for the 32-element optimized Dragon descriptor subset (Dragon32; black points) and the 42-element SMARTS fingerprints (SMARTS42; red points), with glomeruli ordered by Dragon32 performance.

The online version of this article includes the following source data and figure supplement(s) for figure 3:

**Source data 1.** Source data for *Figure 3A, B*.

**Source data 2.** Source data for *Figure 3D-F*.

**Figure supplement 1.** Additional comparisons of glomerular response predictions by physicochemical descriptor sets.

**Figure supplement 1—source data 1.** Source data for *Figure 3—figure supplement 1A-C*.

and methods). Of these, the MACCS feature keys had significantly higher performance metrics than the other descriptor sets (*Figure 3D and E*).

Each of the descriptor sets defined a chemical similarity space using only the 185 odorants in our panel. To evaluate tested odorants relative to their distances within a more general odorant chemical space, as has been done in a recent characterization of odorant coding structure in piriform cortex (*Pashkovski et al., 2020*), we defined a Euclidean chemical space using the first 20 PCs of the matrix of 2982 physicochemical descriptors applied to the curated set of 2624 odorants (*Figure 3—figure supplement 1B*). The projection of odorants in the first two PCs of this space is shown in *Figure 1A*, as in *Pashkovski et al., 2020*. The first 20 PCs captured 66% of the variance in the odorant-descriptor matrix, while the performance metric for predicting effective odorants based on their relative distance within this space reached an asymptote at ~0.6 after 5–10 PCs (*Figure 3—figure supplement 1D*).

It was notable that the chemical space defined by the set of 108 MACCS keys, which have been previously defined to describe general chemical features without regard to odorant-OR interactions, outperformed exhaustive physicochemical descriptor sets and performed equivalently to the 32-descriptor set that was optimized by fitting to odorant-evoked neural response data (*Haddad et al., 2008*). Inspired by this result, we defined a new set of structural features using the SMARTS chemical pattern matching language (Daylight Chemical Information Systems Inc). This set shared some features with the 49-element functional group descriptors ('FG'), with an additional emphasis on resolving larger substructural motifs that are common in odorant compounds, including the presence and length of an aliphatic carbon chain; heteroatom substitutions; ortho-, meta- and para-substituted rings; and a number of terpenoid scaffolds (*Table 2*). The resulting 'SMARTS42' set was small (42 features), but notably had a higher predictive quality than all other descriptor sets, with higher performance metrics across the entire glomerular dataset (*Figure 3E and F*), as well as for the identified glomeruli (*Figure 3—figure supplement 1C*) (SMARTS42 vs. MACCS, p=4.0 × 10⁻⁷, corrected post-hoc Dunn test). These results suggest that, despite the extremely narrow tuning of glomeruli overall, the response spectrum of a given glomerulus in low-concentration regimes is moderately well-predicted from relatively straightforward structural features.

Given this result we hypothesized that, despite the overall high dimensionality of odorant representations arising from narrowly-tuned glomeruli, any correlated relationships among odorant representations would reflect such basic structural features. Indeed, odorant response correlation matrices revealed blocks of correlated responses that corresponded to major structural classes of odorants. In particular, the highest correlations occurred among aliphatic acids, aldehydes and esters; primary amines; and pyrazines and thiazoles/pyrroles/pyridines ('heterocyclic N-S') (*Figure 4A*). As each odorant activated only a few glomeruli, correlation analysis was poorly suited to further investigating the structure of odorant response matrices. Instead, since correlation coefficients essentially reflected co-tuning of individual glomeruli to a few odorants, we constructed co-tuning probability matrices that reflected the likelihood of a glomerulus being responsive to any pair of odorants. As expected, the average co-tuning probability matrix across all eight OBs was very similar to the correlation matrix, with blocks of odorants to which glomeruli are most commonly co-tuned corresponding to major structurally-defined odorant classes, with occasional smaller blocks within a class (*Figure 4B*). To facilitate visualization of odorant co-tuning relationships, we generated a pairwise odorant co-tuning matrix based on the mean number of co-tuned glomeruli per OB, thresholded this matrix at 0.875 co-tuned glomeruli per OB (i.e. co-tuning in at least 7 of 8 OBs), and generated a network graph with odorants as nodes and mean number of co-tuned glomeruli as edge weights (*Figure 5A*).

This visualization confirmed that glomerular tuning was dominated by particular structural relationships between co-tuned odorants. In particular, there was prominent co-tuning between carboxylic acids, aldehydes and esters; between primary amines – including both acyclic and cyclic amines; and between the heterocyclic pyrazines, thiazoles and pyrroles/pyridines (all heterocyclic aromatic compounds containing one or two nitrogens or a nitrogen and sulfur heteroatom). Notably, co-tuning relationships were not well-predicted by the chemical space of odorants defined by computed physicochemical descriptors: for example, the amine odorants were distributed across a large extent of the Good Scents/2982-descriptor space but showed almost no co-tuning to other odorant classes (*Figure 5A and B*). Conversely, the aromatic-containing odorants overlapped within a relatively small extent of descriptor space, but there was little co-tuning between aromatic compounds with different functional groups (*Figure 5A and B*). Other notable aspects of glomerular tuning appeared

**Table 2.** SMARTS42 feature set.

SMARTS42 fingerprints consist of binary keys indicating the presence or absence of each feature. 'SMARTS Pattern' defines each pattern using the SMARTS chemical pattern matching language (Daylight Chemical Information Systems, Inc; https://www.daylight.com/dayhtml/doc/theory/theory. smarts.html).

| # | SMARTS pattern | description |
|---|---|---|
| 1 | *-C(=O)-[OH1] | carboxylic acid |
| 2 | [CH1]=O | aldehyde |
| 3 | C-C(=O)-[O]-C | ester |
| 4 | C-C(=O)-[S]-C | thioester |
| 5 | [!O&!S]-C(=O)-[!O&!S] | ketone |
| 6 | [OX2H][CX4&!$(C([OX2H])[O,S,#7,#15]),c] | alcohol |
| 7 | c1ccccc1 | benzyl |
| 8 | C~C(~C)~C– C~C– C(~C)~C | monoterpene |
| 9 | [#8]1~[#6]~[#6]~[#6]~[#6]1 | furanoid |
| 10 | o1cccc1 | furan |
| 11 | [NH2][C] | primary amine |
| 12 | [NH](C)C | secondary amine |
| 13 | [NH0](C)(C)C | tertiary amine |
| 14 | [N,n]1~[C,c]~[C,c]~[C,c]~[C,c]~[C,c]1 | pyridine |
| 15 | [n,N]1~[C,c]~[C,c]~[C,c]~[C,c]1 | pyrrole |
| 16 | [N,n]1~[C,c]~[C,c]~[N,n]~[C,c]~[C,c]1 | pyrazine |
| 17 | [#16]1~[#6]~[#7]~[#6]~[#6]1 | thiazoline |
| 18 | [!#8]~C S-C~[!#8] | thioether |
| 19 | [$(C-S-S-C),$(C-S-S-S-C)] | sulfide |
| 20 | [#6]-[SH] | thiol |
| 21 | [#6]=[#6] | alkene |
| 22 | [#16] | sulfur |
| 23 | [#7] | nitrogen |
| 24 | [#8] | oxygen |
| 25 | [R] | ring |
| 26 | [CH3]-*-[CH2]-* | 4-bond chain with C at 1 and 3 |
| 27 | *!@*@*!@* | ortho-substituted rings |
| 28 | *!@*@*@*!@* | meta-substituted rings |
| 29 | *1(!@*)@*@*@*(!@*)@*@*@1 | para substituted 6-ring but not fused ring |
| 30 | C~C(~C)~[R1]1~[R1]~[R1]~[R1] (~C)~[R1]~[R1]~1 | menthane scaffold |
| 31 | C~C(~C)~2–[R2]1~[R2]~2–[R1]~[R1] (~C)~[R1]~[R1]~1 | carene scaffold |
| 32 | C~C(~C)~[R2]12~[R1]~[R2]~2–[R1] (~C)~[R1]~[R1]~1 | thujane scaffold |
| 33 | C~C2(~C)~[R1]1~[R]~[R]~2–[R] (~C)~[R]~[R]~1 | pinane scaffold |

*Table 2 continued on next page*

*Table 2 continued*

| # | SMARTS pattern | description |
|---|---|---|
| 34 | [!H]~[!H]2(~[!H])~[R]1~[R]~[R]~[R](~[!H])~2–[R]~[R]~1 | camphane scaffold |
| 35 | [!H]~[!H]2(~[!H])~[R]~[R](~[!H])1~[R]~[R]~2–[R]~[R]~1 | fenchane scaffold |
| 36 | C(-C)(-C)(-C)-C | quaternary carbon |
| 37 | C-C-C-C-C-C | six carbon single bond |
| 38 | C-C-C-C-C-C-C | seven carbon single bond chain |
| 39 | C-C-C-C-C-C-C-C | eight carbon single bond chain |
| 40 | C-C-C-C-C-C-C-C-C | nine carbon single bond chain |
| 41 | C-C-C-C-C-C-C-C-C-C | ten carbon single bond chain |
| 42 | C-C-C-C-C-C-C-C-C-C-C | eleven carbon single bond chain |

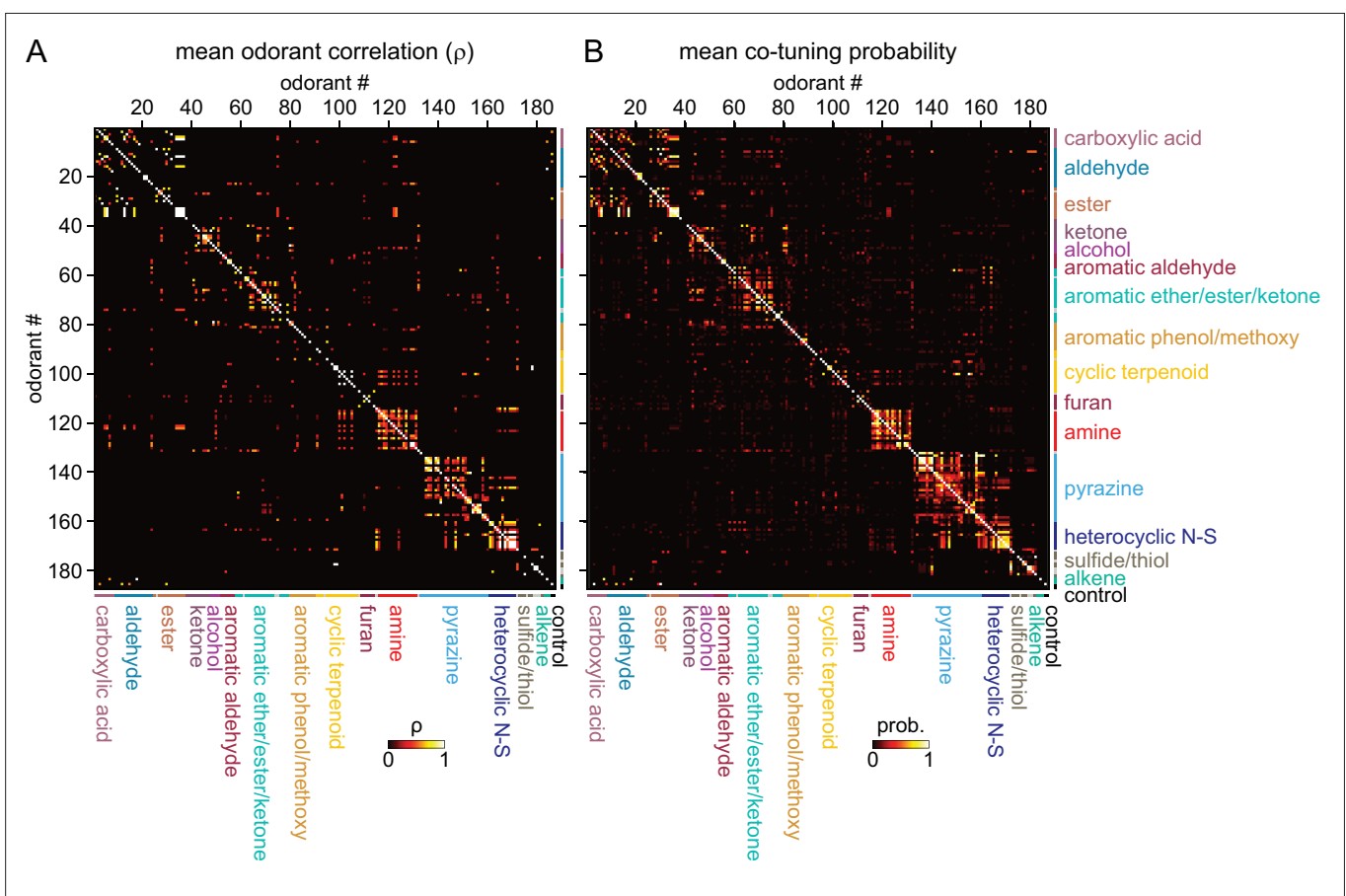

**Figure 4.** Sparse tuning of glomerular inputs is heterogeneously structured. (**A**) Mean glomerular response correlation matrix for the 185-odorant panel. Each value shows the Spearman's rank correlation ($\rho$) between the vectors of glomerular responses evoked by two odorants, averaged across all 8 OBs. Odorants ordered and color-coded according to nominal structural classification (see Materials and methods), as in *Figure 1E*. (**B**) Odorant co-tuning probability matrix for the 185-odorant panel. Each value shows the mean probability of a glomerulus responding to each odorant pair of the matrix, averaged across all responsive glomeruli per OB and then averaged across each of the eight OBs.

The online version of this article includes the following source data for figure 4:

**Source data 1.** Source data for *Figure 4*.

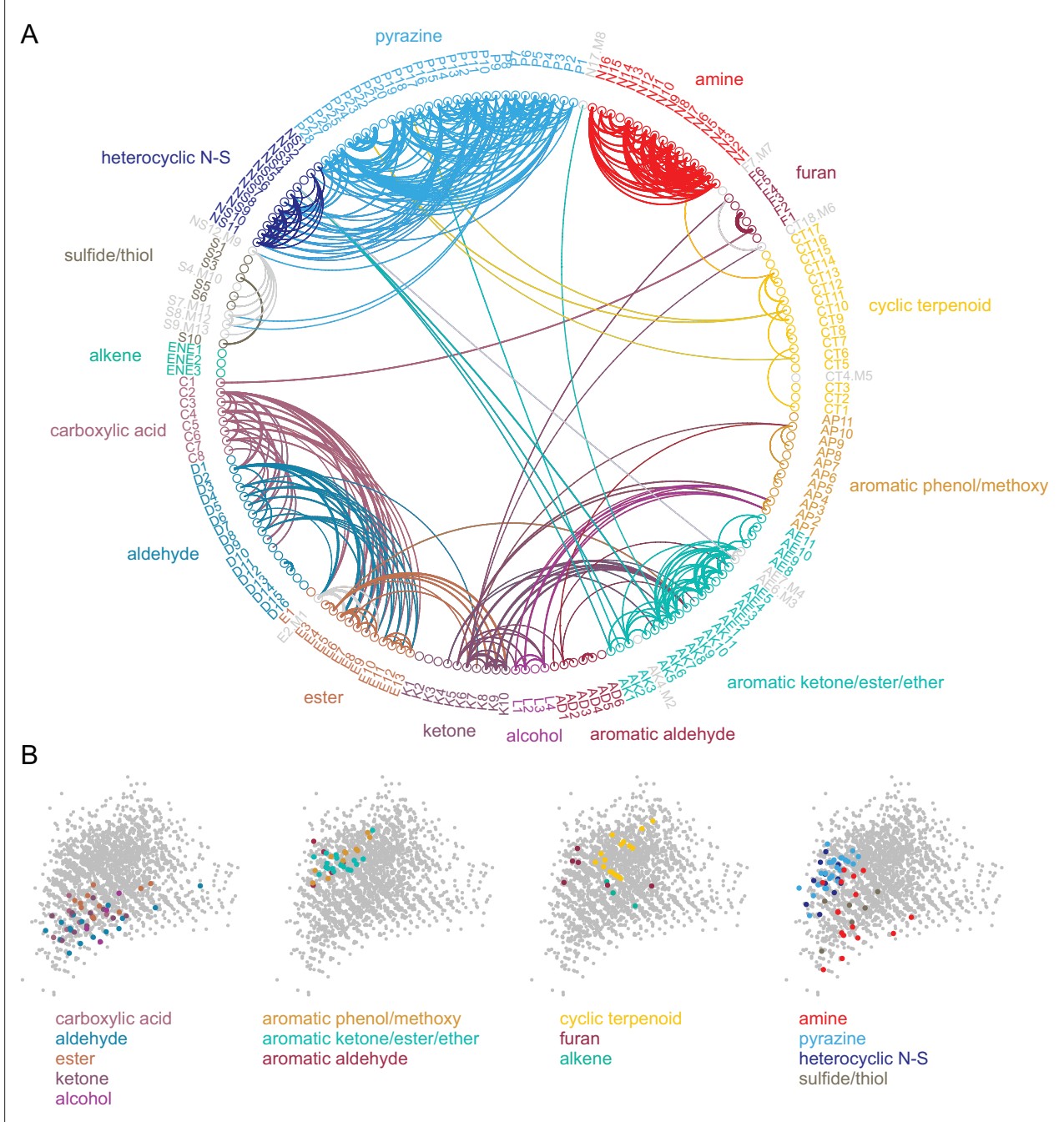

**Figure 5.** Odorant co-tuning relationships reflect basic chemical features of odorants. (**A**) Circular network graph of the most reliable odorant co-tuning relationships, using the mean number of glomeruli co-tuned to each odorant pair. Lines connect odorant pairs with mean co-tuning values above 0.875 (i.e. ≥1 co-tuned glomerulus per OB in at least 7 of 8 OBs). Line thickness scales with co-tuning value. Colors indicate membership in odorant structural group; color of lines connecting odorants across groups chosen to match source group with the most connected members. Letter-number codes indicate odorant identity (**Supplementary file 1**). Odorants with mixed group-defining structural features are shown in grey. Odorants ordered according to nominal structural classification, as in previous figures. (**B**) Odorant panel color-coded by structural group, plotted in the first two PCs of the 2587-odorant physicochemical descriptor space, as in **Figure 1A**. Select odorant classes are highlighted in each replicate plot to facilitate visual comparison. Grey circles indicate all odorants in the database.

The online version of this article includes the following source data for figure 5:

**Source data 1.** Source data for **Figure 5A**.

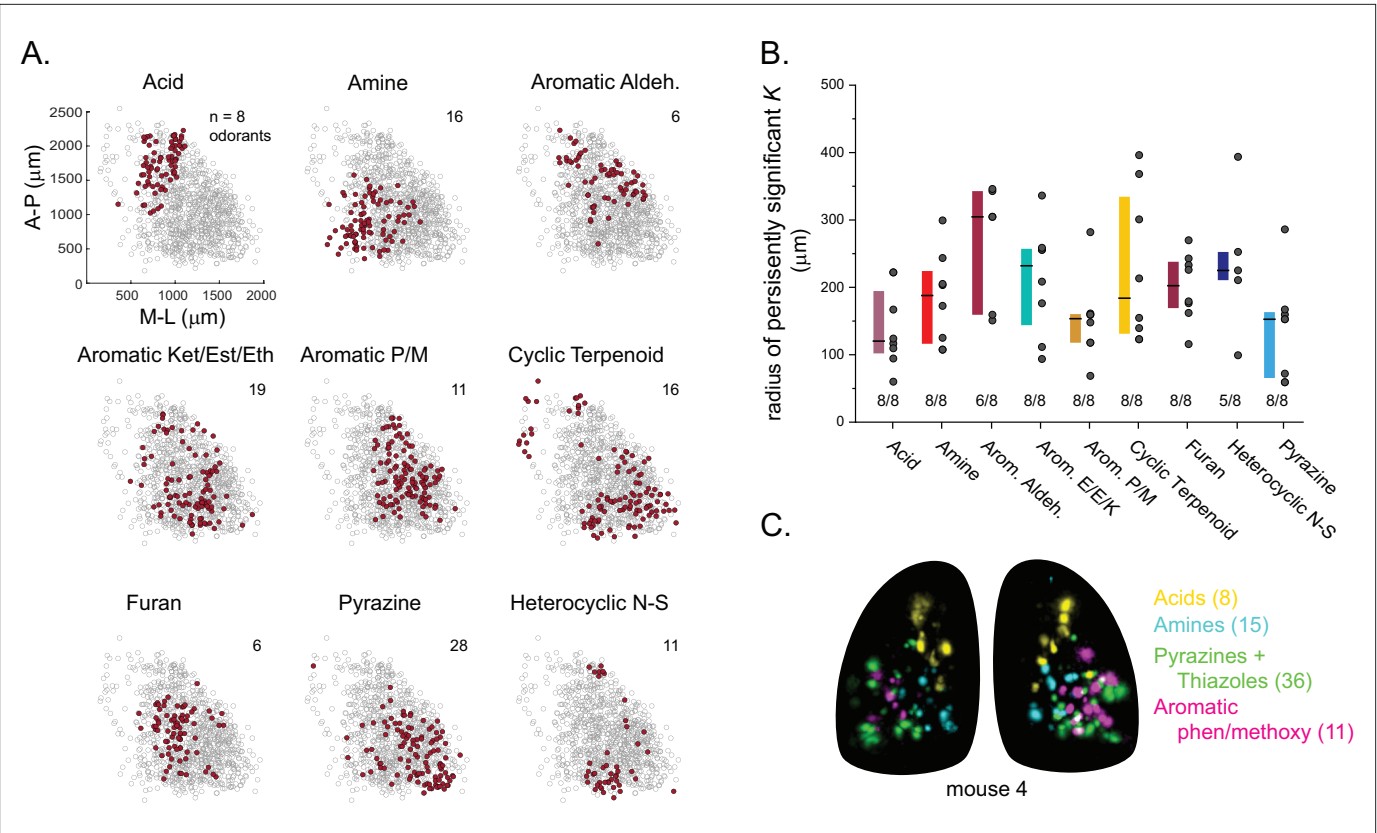

**Figure 6.** Glomerular sensitivity maps reveal spatial clustering of glomeruli tuned to odorant structural classes. (**A**) Glomerular positions across all eight OBs (grey), plotted separately and identified by the structural class of their primary odorant (red). Numbers indicate odorants in each class. (**B**) Size of statistically significant spatial clusters for glomeruli with primary odorants in each odorant class. Radius of persistently significant Ripley's K indicates the minimal radius (in µm) at which the Ripley's K metric remained significant at p<0.01 as radii were progressively increased. Numbers below each bar indicate number of OBs (out of 8) showing statistically significant clustering. Boxes show median and interquartile ranges. Colors match odorant class coloration in previous figures. (**C**) Maximal projection of response maps elicited by odorants within four distinct structural classes: carboxylic acids, amines, pyrazines/thiazoles, and phenol/methoxy-containing aromatics. Numbers indicate number of odorants tested within each class. All data taken from the same mouse. Individual glomeruli show little to no co-tuning to odorants in different classes.

The online version of this article includes the following source data for figure 6:

**Source data 1.** Source data for *Figure 6A*.

**Source data 2.** Source data for *Figure 6B*.

generalizable. For example, co-tuning among the acids, aldehydes and esters was common between odorants with the same or similar carbon chain length (e.g., butyric acid, butyraldehyde, butyric acid esters). However, these glomeruli responded to the acids at 100–1000 x lower concentrations than their aldehyde or ester counterparts (*Supplementary file 1*; *Supplementary file 2*). In addition, while glomeruli sensitive to the sulfur- and nitrogen-containing thiazoles, pyrazines and pyrrole/pyridines showed extensive co-tuning among these classes, they were not sensitive to sulfur- and nitrogen-containing thiols and amines, respectively. Overall, this analysis reveals a basic structure underlying the tuning of the OR repertoire that largely reflects relatively straightforward structural relationships among odorants spanning chemical space.

Finally, we analyzed the spatial organization of glomerular sensitivities with respect to odorant chemical features. Earlier studies have come to varying conclusions about the degree to which chemical features are reflected in the spatial organization of glomerular maps (i.e., 'chemotopy') (*Takahashi et al., 2004*; *Johnson and Leon, 2007*; *Soucy et al., 2009*; *Ma et al., 2012*; *Chae et al., 2019*; *Soelter et al., 2020*). Here, reasoning that any spatial organization in glomerulus tuning should reflect the features of its highest-sensitivity odorants, and building on the finding that glomerular co-tuning largely reflected gross structural features, we first classified odorants based on these features (i.e.

functional group, carbon chain configuration, heteroatom substitution), and then assigned each glomerular position to the structural class of its primary odorant (*Figure 6A*). We then used point-pattern analysis (Ripley's K) to test for a nonrandom spatial distribution of glomeruli tuned to odorants within each class (see Materials and methods). Of the 16 odorant classes considered, nine included sufficient numbers of glomeruli in each OB to support a statistical analysis of spatial organization; of these, seven classes showed significant spatial clustering of glomeruli in every OB; the two remaining classes – aromatic aldehydes and heterocyclic N-S compounds – showed significant clustering in most OBs (6/8 and 5/8, respectively) (*Figure 6A and B*). Individual glomeruli within a cluster were narrowly tuned to only one or a few odorants within a class, indicating that spatial clustering did not arise from overlapping odorant sensitivities among nearby glomeruli. Spatial clustering was expected for the carboxylic acids and amines, which are preferred ligands for class I ORs and TAARs, whose OSN projections selectively target domains in the anterior-medial and central-medial OB, respectively (*Bozza et al., 2009*; *Pacifico et al., 2012*; *Cichy et al., 2019*). Indeed, glomeruli with maximal sensitivities to carboxylic acids and amines showed spatial clustering of glomeruli within these regions (*Figure 6A and C*). For the five remaining odorant classes, clustering was apparent on a spatial scale smaller than the domain defined by the remaining class II OSN projections. For example, pyrazine-sensitive glomeruli were preferentially located in the caudal-lateral most extent of the dorsal OB, while glomeruli sensitive to furans and aromatic compounds were clustered more centrally (*Figure 6A and C*). However, these chemically defined clusters were nevertheless highly overlapping in their spatial extent: glomeruli with distinct odorant sensitivities were spatially interspersed (*Figure 6C*). These results are consistent with a 'mosaic' spatial organization of glomerular chemical sensitivities that has been noted previously for a few odorant classes (*Soucy et al., 2009*; *Chae et al., 2019*), and suggest that such an organization extends across much of olfactory chemical space.

## Discussion

By mapping high-sensitivity odorant responses to OB glomeruli using a large and diverse odorant panel, we found that individual glomeruli – and, by inference, their cognate ORs – are sensitively and selectively tuned to a very narrow portion of odorant chemical space. This narrow tuning leads to a sparse and apparently high-dimensional representation of individual odorants in the regime of low odorant concentrations. At the same time, the small amount of structure that was apparent in this coding regime – both in terms of glomerular tuning and glomerular location – was well-represented by relatively simple descriptions of chemical space derived from odorant structural features.

The prevalence of narrowly-tuned glomeruli, and the degree of selectivity in their tuning, was striking. While some earlier characterizations of OSN tuning have concluded that the majority of OSNs (and, thus, the ORs they express) are narrowly tuned to odorants with shared structural motifs (*Araneda et al., 2000*; *Nara et al., 2011*) – a result qualitatively similar to that seen here – most prior studies have reported a combination of broadly- and narrowly-tuned ORs and OSNs (*Sicard and Holley, 1984*; *Hallem and Carlson, 2006*; *Grosmaitre et al., 2009*; *Saito et al., 2009*; *Nara et al., 2011*; *Del Mármol et al., 2021*), with even those OSNs classified as narrowly tuned appearing less selective than the majority of glomeruli in our dataset. Likewise, some earlier characterizations of mitral/tufted cell tuning in the mouse have indicated high odorant selectivity, but reported lifetime sparseness values across a range of odorants that are substantially lower (indicating broader tuning) than those found here (*Davison and Katz, 2007*; *Fantana et al., 2008*; *Tan et al., 2010*).

A likely explanation for this difference is the much lower range of odorant concentrations used to characterize glomerular specificity here. In general, we presented odorants at concentrations several orders of magnitude lower than those in earlier in vivo studies (*Rubin and Katz, 1999*; *Davison and Katz, 2007*; *Soucy et al., 2009*; *Ma et al., 2012*; *Chae et al., 2019*; *Pashkovski et al., 2020*), and 4–6 orders of magnitude lower than in vitro studies (*Saito et al., 2009*; *Nara et al., 2011*; *Xu et al., 2020*). Glomeruli remained narrowly tuned even when challenged with tenfold higher concentrations, and narrow tuning was not a function of poor signal-to-noise ratios nor of the particular selection of concentrations used for each odorant. Thus, highly selective tuning to odorants appears to be a robust feature of OSN input to OB glomeruli.

The high sensitivity of glomeruli to their primary odorant was also surprising: the median primary odorant concentration across all glomeruli was $2 \times 10^{-11}$ M; threshold concentrations for characteristic odorant-glomerulus pairs are likely even lower, as we did not attempt to determine threshold

sensitivities for any glomerulus. These concentrations are comparable to the sensitivity of the 'ultra-sensitive' TAAR OSNs for their preferred ligands (*Zhang et al., 2013*), and substantially lower than those reported for other OSNs in vivo (*Oka et al., 2006*; *Tan et al., 2010*). This discrepancy could be explained by the smaller odorant panels used in those studies, resulting in a failure to find optimal odorants for a given OSN/OR, as well as a reliance on less-sensitive reporters in detecting responses. We note that the concentrations used here were comparable to or higher than psychophysical detection thresholds measured with rigorous behavioral assays in mice (*Dewan et al., 2018*; *Cichy et al., 2019*; *Williams and Dewan, 2020*), suggesting that the sparse responses at these concentrations can support odor perception.

The narrow tuning of OSNs has implications for how odor information is encoded and processed by central circuits. Canonical models of olfactory information processing rely on a reduced dimensionality of the odorant coding space that arises from systematic overlap in the response spectra of OSN inputs, and predict that central circuits transform odorant representations with respect to this lower dimensional space (*Chae et al., 2019*; *Pashkovski et al., 2020*). Given the low covariance in responses we observed across the OSN population for different odorants, such models may be less applicable in low-concentration regimes. Instead, activation of particular glomeruli may directly encode the presence of a particular odorant, or at least one among a small number of odorants with shared structural features. Notably, sparse and selective tuning to odorant chemical features has been reported earlier for mitral/tufted cells in the mouse, using a large panel of odorants presented at ~1000x - 10,000x higher concentrations (*Davison and Katz, 2007*; *Figure 1—figure supplement 2*); thus, OB circuits may maintain narrow tuning in this higher-concentration regime by suppressing the output of glomeruli that are more weakly activated by their lower affinity odorants (*Yokoi et al., 1995*; *Gire and Schoppa, 2009*; *Cleland and Linster, 2012*).

Natural odors consist of mixtures of many components; in a sparse coding regime, natural odors may elicit combinatorial activity patterns that directly reflect their component composition (*Lin et al., 2006*; *Davison and Katz, 2007*). Narrowly tuned, highly-sensitive glomeruli may also facilitate the recognition of a complex natural odor at trace concentrations based on detection of key components that are signatures of a given odor source (*Dunkel et al., 2014*). Comprehensively defining the odorant tuning features of a large fraction of OB glomeruli, as done here, allows for more precise predictions and stronger tests of how central circuits – for example, in primary olfactory cortex – integrate information across glomeruli to mediate odor object perception.

Despite the overall sparse representation of odorants, we did find systematic relationships in the co-tuning of individual glomeruli to their highest-sensitivity odorants. Co-tuning relationships largely reflected straightforward features of odorant chemical structure; tuning of OSNs or glomeruli to shared structural features has long been noted (*Sicard and Holley, 1984*; *Malnic et al., 1999*; *Rubin and Katz, 1999*; *Wachowiak and Cohen, 2001*; *Takahashi et al., 2004*), although often these relationships have appeared complex due to the relatively broader responsiveness to a range of odorants (*Saito et al., 2009*; *Del Mármol et al., 2021*). The tuning of glomeruli to their highest-sensitivity odorants appears less complex.

Consistent with this organization, models of chemical space that incorporated a relatively small number of odorant structural features were more effective at predicting glomerular co-tuning than models generated from larger sets consisting of exhaustive lists of computationally derived physicochemical properties, and were even more effective than models based on descriptor subsets previously optimized to explain OR-odorant interactions by fitting to response data (*Haddad et al., 2008*). Physicochemical descriptor sets – which are heavily used in computational drug discovery – have been moderately successful at predicting OR ligands and odorant perceptual attributes, but typically require careful tuning of feature selection to training datasets (*Boyle et al., 2013*; *Ravia et al., 2020*; *Gerkin, 2021*; *Kowalewski et al., 2021*) and introduce hazards such as overfitting a large parameter set to a much smaller response dataset (*Chae et al., 2019*). In addition, odorants can differ significantly from drug-like molecules in their size and other chemical features (*Ruddigkeit et al., 2014*). Thus, it is encouraging that the SMARTS42 and MACCS fingerprints used here were equally or more effective at predicting the highest-sensitivity odorants for a given OSN population, as they reflect clear substructural features of odorant molecules and required many fewer descriptors than odorants. These substructure-based models may generalize well across odorant chemical space or across different response measures, and could inform

predictive models of odorant-OR binding (*Poivet et al., 2018*; *Licon et al., 2019*; *Kowalewski and Ray, 2020*) or odor perception.

At the same time, emphasizing the highest-sensitivity odorants for a given OSN population revealed prominent co-tuning relationships among particular odorant groups, indicating a heterogeneous structure to odorant coding space at the level of OSNs that is not adequately captured by a single set of structure-based or computationally derived chemical features. For example, the most extensive co-tuning was observed among glomeruli sensitive to amines, to acids and their corresponding esters and aldehydes, and to pyrazines/thiazoles. The first two groups likely correspond to TAAR and class I OR glomeruli; it will be interesting to determine if the pyrazine/thiazole-tuned glomeruli reflect another distinct subfamily of ORs within the larger class II group. One implication of the heterogeneity of tuning properties across the glomeruli imaged here is that the selectivity of different ORs may be determined by different rule sets. Ultimately, relating OR ligand binding to solved protein structures for mammalian ORs, as has been done recently for an insect OR (*Del Mármol et al., 2021*), will be important in understanding the molecular basis for this heterogeneity.

Structure in odorant representations was also apparent in the spatial organization of glomerular sensitivities with respect to odorant chemical features. We found that spatial clustering of glomeruli with high sensitivities to structurally similar odorants was common, being present in nearly all of the structurally defined classes tested. These results differ from those of earlier studies mapping chemical features across the OB surface, which used higher odorant concentrations and metabolic measures of neural activity and reported relatively discrete spatial clusters of glomeruli responsive to odorants sharing particular chemical features (*Takahashi et al., 2004*; *Johnson and Leon, 2007*). In particular, spatial clustering in our dataset largely arose from the proximal positioning of individual glomeruli sensitively tuned to structurally similar but distinct odorants, rather than from correlated tuning to overlapping sets of odorants or a 'tunotopic' organization (*Ma et al., 2012*). Moreover, with the exception of carboxylic acid-sensitive glomeruli, which were tightly clustered within the presumed domain of class I OSN projections (*Tsuboi et al., 2006*; *Bozza et al., 2009*), spatial clustering of glomerular sensitivities was not discrete, but overlapping and interdigitated. This finding is consistent with the functional mosaic organization of glomerular responses noted previously for aldehydes and thiazoles (*Soucy et al., 2009*; *Chae et al., 2019*), and suggests that this organization is a general feature of glomerular maps with respect to odorant chemical space.

Mapping high-sensitivity odorants to individual glomeruli allowed for the straightforward functional identification of glomeruli across the dorsal OB using only a single diagnostic odorant at a low concentration. The resulting lookup table of glomeruli and their diagnostic odorants provides an efficient means of accessing each of these glomeruli. Previous studies have functionally identified glomeruli using a combination of location and response spectrum across multiple odorants (*Wachowiak and Cohen, 2001*; *Soucy et al., 2009*; *Soelter et al., 2020*), and have used these to link glomeruli to their cognate ORs (*Oka et al., 2006*; *Shirasu et al., 2014*). However, broader use of functionally-identified glomeruli has been limited thus far. Here, we identified at least 26 glomeruli that could be confidently identified - in both anesthetized and awake mice - using only a single diagnostic odorant at a low concentration; we estimate that this number represents approximately 15% of glomeruli on the imaged surface. Testing additional odorants or further reducing odorant concentrations should yield more glomeruli that can be functionally-identified with a single odorant. In the case of glomeruli with sufficiently similar tuning preventing identification using a single odorant, relaxing the strict requirement for singular activation and using additional criteria such as gross position or an additional diagnostic odorant would likely identify several more glomeruli; for example, we identified several amine-sensitive glomeruli that occur as pairs and likely correspond to specific TAAR-expressing OSNs.

We used functionally-identified glomeruli to generate consensus response spectra and to assess variability across OBs and animals; variability was low and likely reflected slight inter-animal differences in physiological state and/or differences in sensitivity or nasal patency. Consensus response spectra were useful in comparing responses to predictions from odorant descriptor sets and in confidently identifying co-tuning relationships that deviated from these predictions. Future studies should be able to probe response spectra for these glomeruli over even larger odorant panels, across concentrations, and chronically in the awake animal. The use of activity reporters with faster kinetics than GCaMP6s (i.e. 'fast' GCaMP variants, iGluSnFRs, or voltage reporters) could further be used to examine the relationship between OR-odorant identity and response dynamics. Functionally-identified glomeruli

should also prove useful for investigating OB circuit transformations using multiplexed imaging from OSN inputs and OB outputs (*Short and Wachowiak, 2019*; *Moran et al., 2021*).

Functionally identifying glomeruli could also facilitate functionally deorphanizing and mapping mammalian ORs across the OB. Indeed, the number of functionally-identified glomeruli from our initial conservative survey exceeds the total number of OR-defined glomeruli whose functional properties have previously been characterized (*Peterlin et al., 2014*; *Shirasu et al., 2014*; *Saito et al., 2017*). Recent spatial transcriptomics efforts have generated maps of estimated glomerular position for ORs across the OB, at a level of spatial precision approaching that of the biological variability in position across animals (*Wang et al., 2022*; *Zhu et al., 2021*). Aligning these maps with the positions of functionally-identified glomeruli should allow candidate ORs for a glomerulus to be narrowed to a few dozen; cross-referencing against in vivo or in vitro functional assays of OR-odorant responsiveness (*Saito et al., 2009*; *Jiang et al., 2015*; *von der Weid et al., 2015*) may then further narrow the list of candidate ORs, accelerating final confirmation through the generation of OR-tagged mice or other approaches such as retrograde labeling of OSNs from the identified glomerulus (*Oka et al., 2006*; *Shirasu et al., 2014*; *Saito et al., 2017*).

The sensitivities – and, potentially, specificity – of glomeruli to particular odorants may change as a function of sampling behavior in awake mice (*Verhagen et al., 2007*; *Eiting and Wachowiak, 2020*; *Jordan et al., 2018*), and may also be shaped by experience: OR expression levels and transcriptional programs that determine odorant responsiveness can change as a function of the recent history of odorant exposure (*Tsukahara et al., 2021*). Aversive conditioning to an odorant can also alter the number of OSNs expressing a given OR and alter the magnitude of OSN responses in specific glomeruli (*Jones et al., 2008*; *Kass et al., 2013*; *Bhattarai et al., 2020*). Here, we found that odorants elicited singular activation of several of the functionally-identified glomeruli at the same picomolar-range concentrations in awake, head-fixed mice as were seen under anesthesia; we also found that the high selectivity of glomeruli to odorants largely persisted over a tenfold increase in odorant concentration. Thus, while absolute sensitivities may be dynamically modulated based on experience and sampling behavior, we predict that response spectra remain consistent and narrowly tuned across a broad range of ethologically relevant contexts. The current datasets and catalog of functionally-identified glomeruli provide a useful platform for testing this prediction in future studies.

An important question for interpreting the ethological significance of the present findings is the range of odorant concentrations encountered by the animal during odor-guided behaviors. Quantitative measurements of vapor-phase concentrations arising from natural odor sources under naturalistic conditions are rare, although it is likely that concentrations vary widely and have a long tail towards lower concentrations as odorants are dispersed by airflow with distance from their source. Even at the source, however, concentrations of many odorants appear to fall within the range of those tested here. For example, ambient concentrations of the most abundant species of volatile organic compounds measured in high-density plant environments range from 0.1 to 10 ppb (*Petersson, 1988*; *Jansen et al., 2009*; *Bach et al., 2020*). Likewise, reports of odorant flux rates from sources such as tomato leaf, forest floor, and freshly cut grass lead to concentration estimates within a similar range when the source is directly sampled via sniffing at its surface (*Ruuskanen et al., 2011*; *Mäki et al., 2019*; *Dehimeche et al., 2021*). Thus, while we have referred to the concentrations used in the present study as low relative to those used in most previous studies, this range likely reflects a common operating regime of the mouse olfactory system during natural behavior.

# Materials and methods

## Key resources table

| Reagent type (species) or resource | Designation | Source or reference | Identifiers | Additional information |
|---|---|---|---|---|
| Genetic reagent (*Mus musculus*, both sexes) | OMP-IRES-tTA | *Yu et al., 2004* PMID:15157418 | RRID:IMSR_JAX:017754 | Mouse line; provided by C. Ron Yu |
| Genetic reagent (*Mus musculus*, both sexes) | tetO-GCaMP6s | *Wekselblatt et al., 2016* PMID:26912600 | RRID:IMSR_JAX:024742 | Mouse line; provided by Jackson Laboratory |

*Continued on next page*

*Continued*

| Reagent type (species) or resource | Designation | Source or reference | Identifiers | Additional information |
|---|---|---|---|---|
| Software, algorithm | custom image analysis GUI | this paper, *WachowiakLab, 2022* | https://github.com/Wachowiak Lab/ImageAnalysisSoftware, (copy archived at swh:1:rev:b40c6f15779fc65b47731d32f375a5f0bf90a64c ) | Matlab scripts |

## Animals

Experiments were performed using both male and female compound heterozygous crosses of OMP-IRES-tTA (Jackson Laboratory stock #017754) (*Yu et al., 2004*) and tetO-GCaMP6s (Jackson Laboratory stock #024742) (*Wekselblatt et al., 2016*) mice aged 2–6 months. Mice were housed up to 5 per cage on a 12 hr light/dark cycle with food and water available ad libitum. All procedures were performed following the National Institutes of Health Guide for the Care and Use of Laboratory Animals and were approved by the University of Utah Institutional Animal Care and Use Committee. (IACUC protocols #19–06007, 19–06008).

## Olfactometry

Odorants were obtained from Sigma-Aldrich, TCI America, Bedoukian Research, or ICN Biomedicals. Liquid dilutions of odorants were prepared to achieve target delivery concentrations of approximately 0.1, 1, 10, 100, or 1000 pM (within an order-of-magnitude) using 1:10 and 1:100 serial dilutions. Non-amine odorants were diluted in caprylic/capric medium chain triglyceride oil (C3465, Spectrum Chemical Mfg. Corp.) within ~1 week of experiments; amine odorants were freshly diluted in water immediately prior to each experiment to minimize odorant oxidation. Trace quantities of Sudan Black B were included in all dilutions to facilitate visual confirmation of olfactometer loading. Diluted odorants were delivered in vapor phase using a custom-built olfactometer equipped with end-stage eductor and operating with 8 L/min charcoal-filtered carrier stream, 30 kPa delivery pressure, 5 cm olfactometer-to-mouse distance, and 2-s long delivery (*Burton et al., 2019*). A fan at the rear of the animal removed odorants after presentation. Odorants were delivered independent of inhalation timing in pseudorandom order, typically in sets of 12 odorants, with 3–5 trials per odorant and 8–10 s inter-trial interval. Eductors were washed with non-scented Alconox detergent and thoroughly rinsed with ethanol and water in between odorant sets to minimize possible odorant adsorption and inter-trial contamination.

## Imaging

Mice were initially anesthetized with intraperitoneal injection of pentobarbital (50 mg/kg) and subcutaneous injection of chlorprothixene (12.5 mg/kg). Subcutaneous injection of atropine (0.5 mg/kg) was further given to minimize mucus secretions and maintain nasal patency. A double tracheotomy was performed as described (*Eiting and Wachowiak, 2018*), and anesthesia was subsequently maintained by ~0.4–0.5% isoflurane delivered in pure $O_2$ to the descending tracheal tube while artificial inhalation (150 ms duration, 300 mL/min flow rate) was continuously driven at 3 Hz through the ascending tracheal tube. Mice were then head-fixed and the bone over the dorsal OB thinned. For widefield imaging experiments, a large well surrounding the dorsal OB was constructed with dental cement and securely covered with a cut glass coverslip, forming a chamber with caudal opening. This chamber was filled with Ringer's solution to render the thinned bone transparent, yielding a cranial window with stable optical plane throughout the experiment. Epifluorescence was collected through a 4×, 0.28 N.A. air objective (Olympus) at 256×256-pixel resolution and 25 Hz frame rate using a back-illuminated CCD camera (NeuroCCD-SM256; RedShirt Imaging) and Neuroplex software, with illumination provided by a 470 nm LED (M470L2, Thorlabs) and green fluorescent protein filter set (GFP-1828A-000, Semrock). For experiments involving both widefield and two-photon imaging trials, epifluorescence was collected similar to above using a 5×, 0.25 N.A. air objective (Olympus), while two-photon fluorescence was collected with 15.2 Hz frame rate using a resonant-scanning microscope (Sutter Instruments) coupled to a pulsed Ti-Sapphire laser (Mai Tai HP, Spectra Physics) tuned to 920 nm and equipped with a 16×, 0.8 N.A. water-dipping objective (Nikon) and GaAsP photomultiplier (Hamamatsu H10770B).

For epifluorescence imaging in awake, head-fixed mice (*Figure 2—figure supplement 1*), the bone overlying both OBs was thinned and sealed with cyanoacrylate (Krazy Glue) to preserve transparency and a headbar was implanted caudal to the OBs (*Wachowiak et al., 2013*). Beginning 3–5 days after the surgical procedure, mice were acclimated to head fixation for periods increasing from 10 to 30 min over 3–4 days. A circular treadmill (design courtesy of D. Rinberg, New York University) allowed for locomotory movements and minimized torque on the headbar. Odorant-evoked responses were imaged on the same optical setup and using the same odorant delivery paradigm as in anesthetized mice. Data were collected over 2–3 consecutive daily sessions lasting 45–60 min.

## Odorant response maps

To generate odorant response maps, raw data from the ≥3 presentations of each odorant were first averaged, then ΔF images generated by, for each pixel, subtracting the mean of the fluorescence signal in the 1 s prior to odorant delivery onset from the mean signal in seconds 2–3 after odorant delivery onset. This time-window nearly always captured the time of the peak of the signal, except in rare cases where responses began to decline before odorant offset, presumably due to adaptation of OSNs. For presentation in the Figures, the resulting ΔF response maps were clipped at zero and a maximum equal to the mean of the highest 65 pixels, smoothed slightly by convolving with a 2D Gaussian kernel (sigma: 0.75 pixels), and the pixel resolution doubled (to 512 × 512 pixels) with bilinear interpolation. Response maps were displayed in units of ΔF as these facilitated visualization of glomerular foci without amplifying noisy signal from low-fluorescence areas such as blood vessels (quantitative analyses were performed after normalization by baseline fluorescence (i.e. on ΔF/F) or by maximal response, as described below).

## Signal extraction and segmentation

For the main dataset ('1 x concentrations', widefield epifluorescence), regions of interests (ROIs) representing single glomeruli were generated using an initial automated selection process followed by manual refinement. First, maximal ΔF projections were generated from the raw (unsmoothed) ΔF response maps across all odorants and initial ROI boundaries generated using 'Find Circles' in Matlab's Image Segmenter App, followed by the 'bwconncomp' function. This initial ROI set was manually refined by addition or adjustment based on visual inspection of the maximal projections and, in some cases, individual odorant response maps. The mean ΔF signal of all pixels from each ROI was used to generate a response matrix (ROI x odorant) for each OB.

Response matrices derived from epifluorescence data required further segmentation to minimize effects from signal spread due to scattered or out-of-focus light arising from nearby glomeruli. Rather than attempt automated segmentation (*Soelter et al., 2014*), we verified that responses were accurately attributed to a given ROI by visual inspection. The inspection procedure involved sorting odorant response maps in descending order of their ΔF response for a given ROI, and setting to zero any entries in the response matrix that did not correspond to clear ΔF signal foci that were centered within the ROI of interest. Signals attributable to responses in adjacent ROIs were easily distinguished with this approach. Due to the sparse nature of odorant responses, the inspection algorithm required visual inspection of, typically, 5–20 response maps per ROI. Inspection was performed blind to odorant identity. In rare cases where odorants evoked clear activity that was not easily segmented into glomerular foci (e.g. octanal response maps, *Supplementary file 2*), no ROI was chosen and these signals were not included for analysis. Finally, responses were normalized to their baseline (pre-odorant) fluorescence to express response magnitudes in terms of relative change (ΔF/F) and to correct for potential differences in GCaMP6s expression level across glomeruli or across mice. A caveat to ΔF/F normalization is that it is unclear whether different baseline F values reflect different expression levels or levels of spontaneous activity, which may vary across ORs and their cognate OSN populations. Thus we provide both normalized and non-normalized response matrices as raw data. This issue is moot for the majority of analyses, which already involve normalizing responses across the odorant panel.

For the 10 x concentration dataset acquired with two-photon imaging, ROIs were selected manually from maximal ΔF projections, and the resulting ΔF/F response matrices were thresholded using a modified z-score cutoff based on the variance in non-responsive odorant trials for each ROI. Variance was calculated as the standard deviation of ΔF/F response values, excluding the top 10th and bottom 5th percentiles of responses across the 185-odorant spectrum. Responses corresponding to

a z-score <9 were set to zero, and the resulting thresholded response matrix was used for further analysis.

## Response matrix statistical measures

To compare response magnitudes across OBs, nonzero ΔF/F values were log-normalized to achieve a normal distribution of magnitude values, then ANOVA was performed on all values grouped by OB. Pairwise post-hoc tests (Tukey's multiple comparison of means) were used to test for differences between OBs.

Lifetime sparseness ($S_L$), a measure of tuning of each glomerulus across the odorant response panel, was calculated as previously (*Davison and Katz, 2007*; *Schlief and Wilson, 2007*; *Pashkovski et al., 2020*). For a set of responses to a single glomerulus across n odorants ($R=\{r_1, r_2, \ldots r_n\}$)

$$S_L = \frac{1}{\left(1 - \frac{1}{n}\right)} * \left[ 1 - \frac{\sum_{i=1}^{n} \left(\frac{r_i}{n}\right)^2}{\sum_{i=1}^{n} \left(\frac{r_i^2}{n}\right)} \right]$$

Similarly, population sparseness ($S_P$), a measure of sparseness of n glomerular responses to each odorant ($G=\{g_1, g_2, \ldots g_n\}$), was calculated as

$$S_P = \frac{1}{\left(1 - \frac{1}{n}\right)} * \left[ 1 - \frac{\sum_{i=1}^{n} \left(\frac{g_i}{n}\right)^2}{\sum_{i=1}^{n} \left(\frac{g_i^2}{n}\right)} \right]$$

For both measures, a value of 1 indicates maximally selective tuning (a glomerular response to only one of the 185 odorants ($S_L$), or a single glomerulus activated by an odorant ($S_P$), while a value of 0 indicates completely nonselective tuning, i.e., equivalent responses to all odorants or equivalent activation of all glomeruli). Paired Wilcoxon signed-ranks tests were used to test for differences in $S_P$ between left and right OB responses to the same odorant presentation.

PCA was performed separately on the response matrix of each OB after normalizing each glomerulus's response spectrum to its maximal response. ED, a measure of the number of PCs required to explain a fraction of the variance in the response dataset, was calculated as described in *Litwin-Kumar et al., 2017*; *Pashkovski et al., 2020*. To compare datasets with different numbers of glomeruli (and, thus, a different maximal number of PCs), we expressed variance explained as a fraction of the total number of PCs.

## Functional identification of glomeruli

To define odorants that could be reliably used to identify putatively cognate glomeruli across OBs, we first identified odorants with reliable singular or near-singular activation of glomeruli using two criteria: no more than two glomeruli activated above a 50% $\Delta F_{max}$ cutoff across all imaged OBs, and activation of at least one glomerulus in at least six of eight OBs, and in each of the four mice. Next, for each potential diagnostic odorant, we identified the glomerulus maximally activated by that odorant in each imaged OB and compared the response spectra of each of these glomeruli across the full 185-odorant panel using Pearson's correlation, taking the median of all pairwise correlations across the 6–8 OBs as a measure of consistency of response spectrum. We also measured the frequency with which the response spectrum of the maximally-activated glomerulus was the most highly correlated with that of the maximally-activated glomerulus in other OBs; this fraction reflected the reliability with which the strongest-responding glomerulus to a given odorant also identified the glomerulus with the most similar response spectrum in another OB. We defined an error ratio as 1 minus this fraction. Odorants were considered diagnostic for functionally-identified glomeruli if they had a median correlation coefficient >0.8 and an error ratio <0.2.

## Odorant classification

To facilitate visual comparison of response maps across the large odorant panel, odorants were nominally grouped according to common structural classifiers including functional group, aliphatic, aromatic or cyclic structure, heteroatom substitution, etc. Odorants with mixed features are identified as such in *Supplementary file 1*. Within each group, odorants were ordered according to progressive

changes in particular features such as carbon chain length. The ordering process was subjective and used only for data visualization. The structural classification was used only for analysis of spatial clustering (e.g. *Figure 6*), as described below.

## Odorant concentrations

Concentrations are reported as estimated concentration delivered to the mouse nose based on the vapor pressure of each odorant, its dilution in liquid solvent, and prior calibrations of the olfactometer using a photoionization detector (*Burton et al., 2019*). We assumed that the odorant and solvent behave as an ideal mixture and that odorants reach saturated vapor concentrations in their delivery reservoir as predicted by Raoult's Law; that is, the odorant and solvent are chemically nonreactive with each other and totally miscible at all proportions. This assumption will not hold across all odorants, but was necessary as data on activity coefficients that describe deviations from ideal behavior are extremely limited for these systems. Furthermore, calculations were based on estimated vapor pressures at 25 °C calculated using the EpiSuite 'mpvpbp' module (US Environmental Protection Agency, https://www.epa.gov/tsca-screening-tools/epi-suitetm-estimation-program-interface) or as reported by The Good Scents company; reported vapor pressures can vary by 30–60% depending on the source and the estimation method. Given these considerations, all reported final concentrations should be considered estimates, and are reported to one significant digit precision (*Supplementary file 1*).

For comparison of concentrations with prior studies (*Figure 1—figure supplement 2*), we compared only odorants in common with our odorant panel. For comparison to *Pashkovski et al., 2020* and (*Davison and Katz, 2007*), we used their reported values of 100 ppm and 10 ppm, respectively. For (*Soucy et al., 2009*; *Chae et al., 2019*) we used a value of 0.5% of saturated vapor (s.v.), based on a midway point of their reported range of 0.05–1% s.v. used in the bulk of experiments. For comparisons to *Ma et al., 2012*, we used the lowest reported effective dilutions of saturated vapor from the 'GIA0512' dataset, available at https://www.pnas.org/doi/full/10.1073/pnas.1117491109#supplementary-materials. The specific odorants compared along with their estimated concentrations are listed in *Supplementary file 3*.

## Chemical descriptor sets

Glomerular response spectra were analyzed with respect to various odorant physicochemical descriptor sets, chosen from previous studies (see Text) and processed as follows: The 'Dragon1377' set was identical to that used in *Haddad et al., 2008*; *Saito et al., 2009*; *Chae et al., 2019*; *Gerkin, 2021* and consisted of an initial set of 1664 descriptors taken from the Dragon database (Kode Systems), 1377 of which showed non-zero variance across our odorant panel. The 'Dragon32' set consisted of the subset of 32 descriptors (a subset of the original 1664) found to optimally predict odorant-OR interactions (*Haddad et al., 2008*). The 'FG' descriptor set consisted of the subset of 154 descriptors defining odorant functional groups (*Saito et al., 2009*), 49 of which had non-zero variance across our odorant panel. The 'Alvadesc2982' set was generated from an expanded list of 5666 descriptors added to the original Dragon database (*Mauri, 2020*), 2982 of which showed non-zero variance across our odorant set. This descriptor set included the same descriptors used in *Pashkovski et al., 2020* (2873 descriptors), with additional descriptors reflecting the larger number of odorants in our panel. For all of the above descriptor sets, descriptor values were calculated using alvadesc (Alvascience, alvaDesc version 2.0.8, https://www.alvascience.com). To normalize for differences in ranges and units across the descriptor set, values for each descriptor across the odorant panel were z-scored, as done previously (*Saito et al., 2009*; *Chae et al., 2019*; *Pashkovski et al., 2020*).

We also defined odorant distances using PCs of the chemical space defined by the 2982 descriptors across 2587 odorants, as performed previously (*Pashkovski et al., 2020*) 2548 odorants were taken from the Good Scents database and met molecular weight criteria for odorous compounds (m.w. between 50 and 300); the remaining 39 were in our odorant panel but not present in this original list and so were added. Parameter values were z-scored as above before performing PCA. The first two PCs, accounting for 39.5% of the total variance, were used to visualize odorants in chemical space (*Figure 1A*). We used the weighting of each odorant across the first 20 PCs, which accounted for 66% of the total variance across the odorant set, to generate a 20 × 185 matrix describing the relative position of each odorant in this reduced chemical space.

Odorant distances were also defined using binary feature keysets representing the presence or absence of a particular chemical feature. 'MACCS' keys (*Durant et al., 2002*), 59 (of 166 total keys) of which were present in our odorant panel, were extracted using alvadesc. The SMARTS42 key set was defined using the SMARTS chemical pattern matching language (Daylight Chemical Information Systems Inc; https://www.daylight.com/dayhtml/doc/theory/theory.smarts.html), as described in the Text. The SMARTS42 set consisted of 42 features (*Table 2*) which were chosen to capture substructures that are present in common-use odorants, including those in our panel (e.g. carboxyl, aldehyde or ketone group; ester bonds; aromatic rings; pyridine substructure; ortho-, meta- and para-substituted rings; a number of terpenoid scaffolds; and the explicit inclusion of single-bond aliphatic chains of lengths from 6 to 11 carbon atoms). In all of the above cases, pairwise (185 × 185) odorant distance matrices were generated for each odorant-descriptor set, using cosine distance as the distance metric for the z-scored physicochemical descriptor sets and [1 - dice similarity] for the binary feature keys.

A performance metric derived from receiver-operating characteristic analysis, an approach commonly used to evaluate the quality of predictive models for receptor-ligand interactions (*Lopes et al., 2017*), was used to evaluate the quality of the different physicochemical descriptor sets as models for predicting glomerular odorant responses. For a given glomerulus, an initial 'query' odorant was defined as the odorant to which that glomerulus responded at the lowest concentration (i.e. primary odorant) or, in some cases (e.g. *Figure 3C*, *Figure 3—figure supplement 1A*) the odorant that elicited the strongest response. The remaining 184 odorants were sorted in order of increasing distance from the query odorant based on a given descriptor set, and the cumulative fraction of all odorant responses explained as a function of odorant rank was plotted, generating a model performance curve, with odorant rank as the false positive rate and fraction of responses explained as the true positive rate. This curve was compared to that derived from the measured responses, generated by sorting odorants in the order of their responsiveness. The performance metric, $P$, was defined as

$$P = 1 - 2 \left( \frac{\text{AUC}_{\text{response}} - \text{AUC}_{\text{model}}}{\text{AUC}_{\text{response}}} \right)$$

where $\text{AUC}_{\text{response}}$ is the area under the cumulative response curve and $\text{AUC}_{\text{model}}$ is the area under the cumulative model performance curve. A model perfectly predicting performance had a value of 1, and chance performance had a value of 0. For measuring model performance for the 19 identified glomeruli with non-singular tuning, median response spectra across all responsive OBs were used, and the query odorant was the primary odorant, with response spectra binarized before evaluation. For comparing performance using the strongest-activating odorant, odorants were sorted in descending order of their response amplitudes and odorant number was plotted against the cumulative fraction of the summed response amplitudes explained. For measuring performance for individual glomeruli across the full dataset, responses were thresholded at 10% of the maximal response for each glomerulus, binarized, and analysis performed only on glomeruli responding above threshold to more than one odorant (694 total glomeruli). For each glomerulus, the performance metric was calculated using each effective odorant as the query odorant (rather than the primary odorant) and the median of these values was reported.

## Response matrix correlation and co-tuning analysis

Odorant response correlation matrices were generated by calculating rank correlation (Spearman's $\rho$) across all glomeruli in a response matrix for all odorant pairs; the resulting odorant correlation matrices were averaged across the eight imaged OBs. Because odorant response vectors typically included only a few responsive glomeruli, we also calculated pairwise odorant co-tuning matrices using a binary measure of responsiveness, where the co-tuning index was equal to the number of glomeruli responsive to a given odorant pair, at any magnitude. Pairwise co-tuning matrices were then averaged across the eight OBs and thresholded at 0.875 (corresponding to an average of at least one co-tuned glomerulus in seven of the eight OBs) to highlight the most consistent odorant co-tuning relationships. Co-tuning relationships were then visualized as a circular network graph using the mean, thresholded co-tuning matrix as the weighted adjacency matrix input. The network graph was rendered using the 'circulargraph' function in Matlab (https://www.mathworks.com/matlabcentral/fileexchange/48576-circulargraph/).

## Spatial clustering analysis

To test for nonrandom spatial distribution of glomerular odorant sensitivities, we used the spatial point pattern measure Ripley's K, which reports whether points within a given radius, r, are dispersed, clustered, or randomly distributed (*Ripley, 1977*). Within each preparation, each glomerulus was assigned to an odorant group based on our structural classification of its primary odorant and Ripley's K was calculated using the set of glomeruli in a given group. Glomeruli with primary odorants having mixed classification (16 of the 185 odorants; see *Supplementary file 1*) were excluded from analysis. Each imaged OB was analyzed separately and treated as a replicate for statistical analysis. A minimum of three glomeruli per structural group in each OB were required to support statistical tests; nine of the 16 groups met these criteria. For these groups, K was calculated for increasing radius (r) values from 0 to ~400 µm (depending on the largest spread of glomeruli in each dataset), in 512 steps of ~0.7–0.8 µm, using the R package 'spatstat' (http://spatstat.org/). Statistical significance of K at each r was calculated by comparing to a Monte Carlo distribution of K values calculated from the same number of glomeruli randomly chosen from the full glomerular array (10,000 iterations). p-values were taken directly from the probability distribution of the Monte Carlo simulation. We used an alpha level of $p < 0.01$ to identify the radius at which nonrandom spatial distributions of glomeruli existed in each dataset; significant Ripley's K values were always greater than the random distribution, indicating spatial clustering. To prevent over-interpreting spurious significant values of r, we further required that Ripley's K remain significant at all increasing radii to consider spatial clustering to be meaningful at a given radius.

## Acknowledgements

We thank Rebecca Kummer, Gustavo Opazo-Vasquez and Alex Price for technical support, Alex Holtmann (University of Utah) for advice on point pattern statistics, Hiro Matsunami (Duke University) for advice on physicochemical descriptors, and Alla Borisyuk and Daniel Zavitz (University of Utah) for help implementing dimensionality measures. We thank C Ron Yu (Stowers Institute, Univ. of Kansas) for providing OMP-IRES-tTA mice and Stan Pashkovski (Harvard Medical School) for providing the list of Good Scents odorants.

## Additional information

### Funding

| Funder | Grant reference number | Author |
|---|---|---|
| National Institutes of Health | NS109979 | Matt Wachowiak |
| National Institutes of Health | MH115448 | Shawn D Burton |
| National Science Foundation | 1555919 | Matt Wachowiak |
| Medical Research Council | 2014217 | Michael Schmuker |
| EU Human Brain Project | SGA3 945539 | Michael Schmuker |
| University of Utah | UROP Award | Audrey Brown |

The funders had no role in study design, data collection and interpretation, or the decision to submit the work for publication.

### Author contributions

Shawn D Burton, Conceptualization, Formal analysis, Investigation, Methodology, Writing - review and editing; Audrey Brown, Thomas P Eiting, Isaac A Youngstrom, Formal analysis; Thomas C Rust, Software; Michael Schmuker, Formal analysis, Methodology; Matt Wachowiak, Conceptualization, Data curation, Formal analysis, Funding acquisition, Investigation, Writing - original draft, Project administration, Writing - review and editing

Author ORCIDs
Shawn D Burton http://orcid.org/0000-0002-8907-6487
Michael Schmuker http://orcid.org/0000-0001-6753-4929
Matt Wachowiak http://orcid.org/0000-0003-4508-9793

Ethics
All procedures were performed following the National Institutes of Health Guide for the Care and Use of Laboratory Animals and were approved by the University of Utah Institutional Animal Care and Use Committee (IACUC protocols #19-06007, 19-06008).

Decision letter and Author response
Decision letter https://doi.org/10.7554/eLife.80470.sa1
Author response https://doi.org/10.7554/eLife.80470.sa2

## Additional files

### Supplementary files

• Supplementary file 1. Table of odorants and estimated concentrations used in the study. Code: abbreviated identifier code, used in *Figure 5A* and *Supplementary file 2*. Epifl. est. conc.: estimated delivered concentration of odorant vapor, in mols/L (M), for 1 x dataset. Most commonly-presented values (of 4 preparations) are shown, reported to one significant digit precision. Dilution: liquid dilution of odorant used to generate delivered concentration. Two-photon rel. conc.: concentration used for the two-photon imaging dataset relative to the concentration used in the widefield epifluorescence dataset. Class: nominal odorant classification based on structural features. Odorants in italics gave no response at the given concentration in any of the 8 OBs.

• Supplementary file 2. Functional Atlas of Sensory Inputs to the Dorsal Mouse Olfactory Bulb. Compilation of olfactory sensory neuron response maps to the 185-odorant panel imaged in each of four OMP-IRES-tTA; tetO-GCaMP6s mice. See Materials and methods for details of odorant presentation, imaging and map generation. Each map is scaled to its own maximum (mean of highest 65 pixels) and clipped at $\Delta F=0$. Concentrations are given as estimated delivered concentration to the mouse nose, rounded to the nearest order of magnitude to allow for deviations from ideal behavior of the odorant in its solvent. See *Supplementary file 1* for more precise estimates of delivered concentration. Grid overlay is for facilitating visual comparison across maps. Odorants are grouped by major structural features and are numbered in order of presentation in Main figures and *Supplementary file 1*. Letter-number identifier matches that in *Figure 5A* and in *Supplementary file 1*. Maps with no measured response in any glomerulus of a given mouse are shown at 50% opacity. Odorants failing to evoke a response in all four mice are indicated with text.

• Supplementary file 3. Table of odorants and estimated odorant concentrations used in prior study comparison. For comparisons to *Ma et al., 2012* we used the lowest reported effective dilutions of saturated vapor from the 'GIA0512' dataset, available at https://www.pnas.org/doi/full/10.1073/pnas.1117491109#supplementary-materials. Estimated molar concentrations were calculated based on reported vapor pressures, as described in the Materials and methods.

• Supplementary file 4. Atlas of Functionally Identified Glomeruli. Compilation of diagnostic odorants and example response maps for each of the 26 functionally-identified glomeruli (see Text and *Table 1*). Concentrations are given as estimated delivered concentration, rounded to the nearest order of magnitude. See *Supplementary file 1* for more precise estimates of delivered concentration. Bar plots indicate relative response magnitudes evoked by all effective odorants for each glomerulus, taken from the median response spectrum across the eight OBs. Positions of each glomerulus are shown in lower right of p. 1, reproduced from *Figure 2*.

• Supplementary file 5. Additional odorants and concentrations eliciting consistently sparse activation but failing conservative requirements for functional identification. See Materials and methods and *Table 1* for definition of Error ratio and Median ORS corr.

• MDAR checklist

### Data availability

Data and code underlying all analyses, including response matrices for all imaged OBs and GUIs for interactively visualizing the datasets, are available from https://github.com/WachowiakLab, (copy archived at swh:1:rev:b40c6f15779fc65b47731d32f375a5f0bf90a64c) and in the provided Source Data files.

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
