## [Editor Report]

This paper investigates how odors are represented in the olfactory bulb of the brain. Classical studies have revealed a 'combinatorial code' for odorant recognition, with individual odorants represented by combinations of broadly tuned and low affinity olfactory receptors. Here, the authors perform a large scale analysis of odor responses across glomeruli, and surprisingly observe that odorant receptors instead generally display remarkably narrow tuning profiles.

---

## [Decision Letter]

**Decision letter after peer review:**

Thank you for submitting your article "Mapping odorant sensitivities reveals a sparse but structured representation of olfactory chemical space by sensory input to the mouse olfactory bulb" for consideration by *eLife*. Your article has been reviewed by 3 peer reviewers, including Stephen Liberles as the Reviewing Editor and Reviewer #1, and the evaluation has been overseen by a Reviewing Editor and Piali Sengupta as the Senior Editor. The following individual involved in the review of your submission has agreed to reveal their identity: Bettina Malnic (Reviewer #2).

Essential revisions:

The reviewers were enthusiastic about the manuscript, but there were a few requests to solidify the data. We have included the full reviewer comments below, and encourage you to focus your revision on a few items.

1) In preparing the revision, it was encouraged to focus on resolving technical questions related to the imaging preparation and associated interpretations. This pertains to the major comments of reviewer #3 and the second and third comments of reviewer #2.

2) We recognize that it may not be feasible to link glomerulus identity to OR identity. Any information that can be provided without additional experiments would strengthen the manuscript considerably but is not a requirement for publication.

*Reviewer #2 (Recommendations for the authors):*

Many Figure legends are incomplete and do not describe well the Figure for the general reader. For example:

Figure 2B – The name of the odorants depicted in the figure should be described in the Figure legend.

Table S2 – Explain in the legend what the anterior-posterior and mediolateral positions mean.

Citation of some primary work is missing, for example:

Benjamin D. Rubin, Lawrence C. Katz, Optical Imaging of Odorant Representations in the Mammalian Olfactory Bulb, Neuron, Volume 23, Issue 3, 1999.

https://doi.org/10.1016/S0896-6273(00)80803-X

This is a work in rats, but one of the first to visualize the functional responses in the bulb in living animals, to show that glomeruli were tuned to detect particular molecular features and that maps of similar molecules were highly correlated.

Gilles Sicard, Andre´ Holley, Receptor cell responses to odorants: Similarities and differences among odorants, Brain Research, Volume 292, Issue 2, 1984, Pages 283-296,

https://doi.org/10.1016/0006-8993(84)90764-9

This is in frogs, but one of the first comparing odorant structural features in single olfactory neuron responses.

*Reviewer #3 (Recommendations for the authors):*

In "Mapping odorant sensitivities reveals a sparse but structured representation of olfactory chemical space by sensory input to the mouse olfactory bulb", Burton et al., aim to functionally map odorant responses across the olfactory bulb dorsal surface. The authors characterized glomerular responses to a large odorant panel at low concentration ranges and derived a robust map of 25 glomeruli that are sensitive to a primary odorant. This manuscript represents an important resource to the community, however, there are some points of interpretation/analysis that require revision.

1. In the current analyses, the rationale for not fold-normalizing to the baseline is not adequately provided (given biological heterogeneity e.g., reporter expression); importantly, the δ F alone could be misleading in subsequent PCA analyses (lines 607-608). We recommend converting to a dF/F in this report.

2. The main observation that the authors emphasize is the high dimensionality of the glomerular responses to a large panel of low-concentration odorants. There are a couple of important points here. First, it is not surprising that at low concentrations dimensionality should rise, given what we know about how GPCRs work – in this sense, it is perhaps best to state this as an obvious prediction that is borne out by the data. Second, the dimensionality of neural responses to odors is the consequence of two separate things: odor concentration and the set of odors tested. The authors repeatedly point out that what they are observing is higher dimensional than what has been observed by others, despite the fact that the data being compared are nearly always querying different parts of odor space at different levels of resolution. Some clarity over this point as in the results and Discussion section would be useful – and perhaps most useful would be apples-to-apples comparisons where these authors have looked at the same odors as others. In the absence of that, it is difficult to disentangle whether the reason for the high dimensionality is because of concentration or differences in odor identity across the odor set.

3. Lines 697-700: The authors assessed how well each of these odorant-descriptor distances predict the glomerular responses. In the methods, the "model" part of this analysis is not clearly described – it seems like the authors are asking how, across glomeruli and odors, the rank order of glomerular activity observed relates to the rank order of odor distances given an odor distance metric (quantified using an auROC). If I'm understanding this correctly, this is not a "model" in the sense that nothing is fit, nothing is held out, and there is no statistical metric generated that shows the "model" output is not a consequence of chance. I'm actually cool with this way of capturing neural activity-odor distance relationships, but more description and perhaps a change in language are important for clarity. More importantly, however, the author seems to be looking at cosine distances of each descriptor set and then judging descriptor quality, despite the fact that each of these descriptor sets has its own covariance structure and thus variation in information about odor relationships is differentially distributed across dimensions. The authors mention using 20 PCs (which account for some amount of variance) for one of the odor descriptor sets. The key question here is: given an equal amount of variance captured, how do odor descriptor sets compare? It is not possible to know that answer without PCA being applied to all the descriptor sets (in the same manner that it was for that one odor set) and to then look at performance a. as a function of PCs and b. as a function of variance explained; if the 42 element SMARTS set outperforms DRAGON given an equal amount of chemical variance explained, it would mean that SMARTS is fundamentally more informative than DRAGON. Absent this kind of normalization it is hard to know why one odor descriptor set outperforms another.

---

## [Author Response]

Essential revisions:Reviewer #2 (Recommendations for the authors):Many Figure legends are incomplete and do not describe well the Figure for the general reader. For example:Figure 2B – The name of the odorants depicted in the figure should be described in the Figure legend.

We have added the letter-number codes to indicate odorant identity, as listed in Supplementary File, directly on the Figure panel.

Table S2 – Explain in the legend what the anterior-posterior and mediolateral positions mean.

We have added this to the Figure legend. We have additionally revised other figure legends throughout the manuscript to more completely and clearly describe the figures.

Citation of some primary work is missing, for example:Benjamin D. Rubin, Lawrence C. Katz, Optical Imaging of Odorant Representations in the Mammalian Olfactory Bulb, Neuron, Volume 23, Issue 3, 1999.https://doi.org/10.1016/S0896-6273(00)80803-XThis is a work in rats, but one of the first to visualize the functional responses in the bulb in living animals, to show that glomeruli were tuned to detect particular molecular features and that maps of similar molecules were highly correlated.Gilles Sicard, Andre´ Holley, Receptor cell responses to odorants: Similarities and differences among odorants, Brain Research, Volume 292, Issue 2, 1984, Pages 283-296,https://doi.org/10.1016/0006-8993(84)90764-9This is in frogs, but one of the first comparing odorant structural features in single olfactory neuron responses.

We appreciate these suggestions and have incorporated these citations into the main text in several places (Introduction, line 39; Discussion, lines 443, 452).

Reviewer #3 (Recommendations for the authors):In "Mapping odorant sensitivities reveals a sparse but structured representation of olfactory chemical space by sensory input to the mouse olfactory bulb", Burton et al., aim to functionally map odorant responses across the olfactory bulb dorsal surface. The authors characterized glomerular responses to a large odorant panel at low concentration ranges and derived a robust map of 25 glomeruli that are sensitive to a primary odorant. This manuscript represents an important resource to the community, however, there are some points of interpretation/analysis that require revision.1. In the current analyses, the rationale for not fold-normalizing to the baseline is not adequately provided (given biological heterogeneity e.g., reporter expression); importantly, the δ F alone could be misleading in subsequent PCA analyses (lines 607-608). We recommend converting to a dF/F in this report.

The majority of analyses (including the PCA analysis) involve normalizing the response spectra of each glomerulus across the odorant panel, such that using dF alone or dividing by resting F (for dF/F) should yield equivalent results. We do understand this concern, however, and have recalculated the other analyses – including the sparseness and effective dimensionality measurements – using dF/F, with near-identical results. We have also revised the relevant text in the Methods, with rationale for using dF/F (lines 707 – 713).

2. The main observation that the authors emphasize is the high dimensionality of the glomerular responses to a large panel of low-concentration odorants. There are a couple of important points here. First, it is not surprising that at low concentrations dimensionality should rise, given what we know about how GPCRs work – in this sense, it is perhaps best to state this as an obvious prediction that is borne out by the data. Second, the dimensionality of neural responses to odors is the consequence of two separate things: odor concentration and the set of odors tested. The authors repeatedly point out that what they are observing is higher dimensional than what has been observed by others, despite the fact that the data being compared are nearly always querying different parts of odor space at different levels of resolution. Some clarity over this point as in the results and Discussion section would be useful – and perhaps most useful would be apples-to-apples comparisons where these authors have looked at the same odors as others. In the absence of that, it is difficult to disentangle whether the reason for the high dimensionality is because of concentration or differences in odor identity across the odor set.

To respond to the first point, we agree that dimensionality should decrease at lower concentration, simply on first principles arising from receptor tuning; however, the degree to which this should occur is unclear, as dimensionality depends not only on the specificity of single receptors but also on the degree of overlap in the specificities of different ORs/glomeruli. Thus, it seems important to characterize this, and the high degree of selectivity and low degree of overlap in tuning (which together increase dimensionality) remains surprising, in our opinion, in light of the bulk of prior literature characterizing these features. We have reviewed our presentation of these results, and have made minor edits to the text to be sure we cite prior work highlighting high sparseness and the impact of odorant concentrations on tuning (in particular, Davison and Katz, J Neurosci 2007, and Meister and Bonhoeffer, J Neurosci 2001). (e.g., Discussion, line 448; lines 479 – 484).

To the second point, we agree that differences in the concentrations and composition of odorant sets are a pervasive problem of our field that hinders comparisons of basic response properties. We do note that the composition of the odorant panel in our study appears to cover a similar portion of chemical space – and include largely the same odorant classes – as those used in the prior studies that we cite, with the exception of pyrazines, of which we have a large number and which are absent or underrepresented in most prior mammalian studies.

Nonetheless, to begin to address this concern, as suggested by the reviewer we have taken a subset of odorants that are identical between our study and that of Ma et al. (PNAS 2012; https://www.pnas.org/doi/full/10.1073/pnas.1117491109); this is an advantageous comparison since that study also imaged GCaMP signals from OSNs using a near-identical approach, including the same OMP-tTA driver line (albeit with GCaMP2), and the authors have shared their raw response matrices online, allowing us to make direct quantitative comparisons. There were 31 odorants in common between the two studies; we extracted their shared response data using the lowest effective concentrations tested in their experiments (identical to the concentration comparison we show in Figure 1 —figure supplement 2). We then compared measures of selectivity, sparseness and dimensionality to those from response matrices made using only these 31 odorants from our datasets. As expected, we found that glomeruli tested in the Ma et al. study with the same odorants at ~1000x higher concentration were less selective, odorant-evoked responses less sparse, and response patterns across the odorant panel had lower dimensionality. These results are reported in a new paragraph (lines 191 – 210); the list of odorants compared (and their concentrations) is included as Supplementary file 3.

3. Lines 697-700: The authors assessed how well each of these odorant-descriptor distances predict the glomerular responses. In the methods, the "model" part of this analysis is not clearly described – it seems like the authors are asking how, across glomeruli and odors, the rank order of glomerular activity observed relates to the rank order of odor distances given an odor distance metric (quantified using an auROC). If I'm understanding this correctly, this is not a "model" in the sense that nothing is fit, nothing is held out, and there is no statistical metric generated that shows the "model" output is not a consequence of chance. I'm actually cool with this way of capturing neural activity-odor distance relationships, but more description and perhaps a change in language are important for clarity. More importantly, however, the author seems to be looking at cosine distances of each descriptor set and then judging descriptor quality, despite the fact that each of these descriptor sets has its own covariance structure and thus variation in information about odor relationships is differentially distributed across dimensions. The authors mention using 20 PCs (which account for some amount of variance) for one of the odor descriptor sets. The key question here is: given an equal amount of variance captured, how do odor descriptor sets compare? It is not possible to know that answer without PCA being applied to all the descriptor sets (in the same manner that it was for that one odor set) and to then look at performance a. as a function of PCs and b. as a function of variance explained; if the 42 element SMARTS set outperforms DRAGON given an equal amount of chemical variance explained, it would mean that SMARTS is fundamentally more informative than DRAGON. Absent this kind of normalization it is hard to know why one odor descriptor set outperforms another.

We believe that the concerns here arise largely from an inadequate explanation/justification of our choice and evaluation of the chemical descriptor sets. We used the term ‘model’ in the sense of a simplification of a complex entity for the purposes of further exploration – in this case, the complex entity is the chemical space of odorants, and the various choices of chemical descriptors attempt to define different (necessarily simplified) models of that space. We agree that this terminology can imply a goal of adjusting/fitting the model to optimally fit our results, which we did not do, so we have minimized and clarified the use of this term. We also have revised the text to hopefully clarify our approach and to clarify the rationale for the choice of the different descriptor sets (lines 313 – 338). This includes clarifying that our auROC performance metric is a standard approach used in ligand-based virtual screening for evaluating the predictive quality of different structural determinants of ligand-receptor interactions.

As to the issue of comparing performance across our different descriptor sets as a function of variance explained, and using only 20 PCs of the GoodScents descriptor set versus the full descriptor sets in the other cases, we believe this concern is also addressed with clarification. The key distinction between the ‘GS20’ set and the other sets is that all other sets define a chemical space based on descriptor distances between *only the odorants in our panel,* whereas the GoodScents descriptor set actually describes a different chemical space – one that encompasses thousands of odorants (the 2624 odorants from the GoodScents database) and uses a larger number of descriptors (2982 descriptors). The rationale for evaluating this space was to evaluate how glomerular tuning relates to odorant similarities *within this more general space of odorants*, not only those tested in our study (which is subject to some form of selection bias). We did not adequately explain this in the original manuscript.

To perform this comparison (i.e., to project our odorants into the chemical ‘space’ defined across the GoodScents odors), it was necessary to use a matrix of principal component weights derived from this 2624 odorant x 2982 descriptor matrix, applied to each of our tested odorants. Using PCA dimensionality reduction allows for a simplified definition of this large chemical space that we can then apply to our smaller odorant panel, preserving the most distinctive physicochemical relationships between odorants. We note that using all of the PCs (describing 100% of the variance of the GoodScents odor descriptor space) reduces to simply applying all of the 2982 descriptor values, unweighted, to our odorants – we actually did do this, and reported this performance metric for the functionally-identified glomeruli in the original submission (currently reported on line 324). Not surprisingly, this descriptor set performed similarly to the 1377-element Dragon descriptor set (the 2982-element GoodScents set is simply an expanded version of the Dragon set).

While we chose 20 PCs somewhat arbitrarily (a point in which performance has clearly reached an asymptote), we also showed in Supplementary Figure S7 (now Figure 3 —figure supplement 1D) the performance metric (and variance explained) as a function of PC#, as the Reviewer asks.

We have revised the text and the order of presentation of the performance metrics for the different descriptor sets to clarify this issue and better explain the rationale for using the ‘GS20’ set (lines 339 – 347).